



# The discontinuous Galerkin coastal and estuarine modelling system (DGCEMS v1.0.0): a three-dimensional, mode-nonsplit, implicit-explicit Runge–Kutta hydrostatic model

Zereng Chen[1,2,3], Qinghe Zhang[1], Guoquan Ran[4], and Yang Nie[1]

[1] State Key Laboratory of Hydraulic Engineering Intelligent Construction and Operation, Tianjin University, Tianjin 300350, China
[2] The National Key Laboratory of Water Disaster Prevention, Hohai University, Nanjing, 210024, China
[3] College of Harbour, Coastal and Offshore Engineering, Hohai University, Nanjing, 210024, China
[4] School of River and Ocean Engineering, Chongqing Jiaotong University, Chongqing 400074, China

*Correspondence to*: Qinghe Zhang (qhzhang@tju.edu.cn), Guoquan Ran (2015205064@tju.edu.cn)

**Abstract.** Numerical method of discontinuous Galerkin (DG) discretization for coastal ocean modelling have advanced significantly, but there are still challenges in accurately simulating phenomena such as wetting and drying process and baroclinic flows in coastal and estuarine regions. This study develops a novel 3D coastal and estuarine modelling system, DGCEMS, using a quadrature-free nodal DG method. The model adopts $\sigma$-coordinates, employs a non-split mode

framework, and integrates a semi-implicit Runge–Kutta scheme with second-order accuracy in both space and time. A series of numerical experiments demonstrate the model's second-order convergence, low spurious mixing, and capability to simulate salt-freshwater interactions in the presence of wetting and drying boundaries.

## 1 Introduction

During the past few decades, numerical methods have dramatically evolved, and coastal ocean models can benefit from these
advanced numerical methods (Blaise et al. 2010). Classical three-dimensional hydrodynamic models, e.g. ROMS (Shchepetkin and McWilliams, 2005), Delft3D (Lesser et al., 2004), FVCOM (Chen et al., 2003), TELEMAC (Moulinec et al., 2011) and SCHISM (Zhang et al., 2016), can be utilized to predict storm surges, tsunamis, floods, and assess the environmental qualities by adding additional equations (e.g., transport, reaction), to model oil slicks, contaminant plume propagation, temperature and salinity transport, among other problems (Bertsch et al., 2022; Zhou et al., 2024; Sanz-Ramos
et al., 2024).

In coastal and estuary waters, the presence of large gradient and strong discontinuity processes, (e.g. wetting and drying process and saline flow), may bring some challenges for these classical circulation models, such as maintaining high numerical accuracy, minimizing numerical mixing, and preventing excessive diffusion. From the perspective of numerical discretization methods in the above models, the finite volume (FV) method is known to have limitations in developing high-
order accuracy schemes and incorporating so-called *hp*-adaptivity (Lee, 2019). High-order FV models can increase the order



by wide stencils but they have to perform complex numerical reconstruction which significantly increases computational complexity (Cheng et al., 2016). Models based on the finite element (FE) method can also achieve high order while it does not have local conservation. In the wet-dry (WD) fronts and small-scale dynamics like baroclinic eddies, where the local conservations of mass and momentum are critical, the property of local conservation is a preferred characteristic of a numerical scheme (Lee, 2020).

As a combination of both FV and FE methods, the discontinuous Galerkin (DG) method is well suited to the modelling, with a relatively small number of elements, of three-dimensional (3D) flows exhibiting strong velocity or density gradients. Dawson and Aizinger (2005) first developed a three-dimensional (3D) shallow water equation (SWE) model based on DG methods called UTBEST3D. They used $z$-coordinates in a vertical direction and a local discontinuous Galerkin (LDG) method for vertical diffusion (Aizinger and Dawson, 2007). To reduce the degrees of stiffness, Blaise et al. (2010) and Comblen et al. (2010) developed a 3D baroclinic marine model called SLIM3D and used symmetric interior penalty Galerkin method (SIPG) to discrete the vertical diffusion. Besides, the mode-splitting procedure, which splits the 2D external mode and the 3D internal mode, is used to save computational time (Kowalik and Murty, 1993). Later, Delandmeter et al. (2018) added a fully consistent and conservative vertically adaptive coordinate system in SLIM3D. At the same time,

Kärnä et al. (2018) also developed a 3D DG baroclinic model called Thetis. Thetis includes a more accurate mode splitting method, revised viscosity formulation, and new second-order time integration scheme, which has lower or comparable numerical dissipation. The above three models all choose $z$-coordinates. Considering that $\sigma$-coordinates enable a smooth representation of the bottom topography, which is particularly appropriate for coastal applications. Using $\sigma$-coordinates, Colton and Kubatko (2016) explicitly treated the vertical diffusion term to study the water column concerning barotropic

forcing by building a 3D DG SWE model. The model of Colton and Kubatko (2016) did not consider the non-linear advective terms, and can only investigate 3D linear problems. The mentioned models use the quadrature-based nodal DG method. In explicit and semi-implicit time stepping schemes commonly used in connection with the SWE models, the most computationally expensive parts of a DG algorithm are the element and edge integrals computed via loops over quadrature points (Faghih-Naini, 2020). One criticism of the original DG method is that it requires expensive quadrature rules to reduce

the aliasing error arising in the nonlinear system. The essence of the quadrature-free approach is to convert the nonlinear fluxes to polynomials or polynomial operations, which can be integrated analytically into the element or element boundaries (Li, 2024). The quadrature-free DG scheme has comparable order and $L_2$ errors even at the 7th order of accuracy compared to the quadrature-based DG scheme for the isentropic vortex problem (Nair, 2015). In the Matlab-C hybrid framework, Ran et al. (2023) developed a 3D barotropic model based on quadrature-free nodal DG method. They choose the mode non-split

scheme considering that this method does not need to adjust the differences in the momentum calculation between the two-dimensional external mode and the three-dimensional internal mode, and has fewer excessive errors (Chen et al., 2022; Colton and Kubatko, 2016). A series of numerical experiments have proven the accuracy and potential application value of the barotropic model, but its computational efficiency is low which limits the applicability of the model.





In addition to the utilization of different numerical discretization scheme, the wetting and drying (WD) process should be
included in the coastal and estuarine modelling, which is a challenge for 3D DG model. At present, within existing 3D DG
models, Vallaeys (2018) noted that the WD process can be considered in the external mode of the mode-splitting SLIM3D
model, but this kind of wetting and drying module cannot be operated together with the baroclinic module. Chen et al. (2024)
developed a WD treatment in a 3D DG model with mode non-split scheme, but there is still no description and verification
of the coexistence of WD processes and thermohaline transport.

To adapt to the complex terrain of the estuary and coastal areas and have a comparable calculation accuracy to the existing
3D DG baroclinic models, this work aims to develop a new share-code discontinuous Galerkin coastal and estuarine
modelling system (DGCEMS) for 3D mode non-split, implicit-explicit Runge–Kutta baroclinic model using σ-coordinates
with a WD module based on quadrature-free nodal DG scheme. The arrangement of this paper is as follows. Section 2
describes the model with discretization of each term in the model, while section 3 gives numerical tests and applications to
show the main functionality of the model. Conclusions are presented in Section 4.

## 2 Model and discretization

### 2.1 Governing equations

The conservative and pre-balanced scheme of 3D shallow water equations in the $\sigma$ coordinates, with advection-diffusion
equations of temperature $T$ and salinity $S$, can be written as:

$$\mathbf{U}_t + \nabla \cdot \mathbf{F}(\mathbf{U}) = \mathbf{S}(\mathbf{U}) \tag{1}$$

where $\mathbf{U} = [D, Du, Dv, DT, DS]^{\mathrm{T}}$; $D$ is the water depth; $u$ and $v$ are the velocity components along the $x$ and $y$ directions,
respectively; the convective term $\mathbf{F}(\mathbf{U}) = [\mathbf{E}(\mathbf{U}), \mathbf{G}(\mathbf{U}), \mathbf{H}(\mathbf{U})]$ and each part in this term can be written as:

$$\mathbf{E}(\mathbf{U}) = \begin{bmatrix} Du \\ Du^2 + 1/2\left(g\left(D^2 - z_b^2\right)\right) \\ Duv \\ DuT \\ DuS \end{bmatrix}, \quad \mathbf{G}(\mathbf{U}) = \begin{bmatrix} Dv \\ Duv \\ Dv^2 + 1/2\left(g\left(D^2 - z_b^2\right)\right) \\ DvT \\ DvS \end{bmatrix}, \quad \mathbf{H}(\mathbf{U}) = \begin{bmatrix} \omega \\ \omega u \\ \omega v \\ \omega T \\ \omega S \end{bmatrix} \tag{2}$$

where $g$ is gravitational acceleration; $z_b$ is the bottom elevation and $\omega$ is vertical velocity along the $\sigma$ direction. The right
side of Eq. (1) contains the terms of:

$$\mathbf{S}(\mathbf{U}) = \mathbf{S}_b + \mathbf{S}_f + \mathbf{S}_{d,h} + \mathbf{S}_{d,v} + \mathbf{S}_{baro} \tag{3}$$

where $\mathbf{S}_b$, $\mathbf{S}_f$, $\mathbf{S}_{d,h}$, $\mathbf{S}_{d,v}$ and $\mathbf{S}_{baro}$ represent the bottom topography term, Coriolis acceleration term, horizontal diffusion
term, vertical diffusion term, and baroclinic term, respectively. The five terms are given as follows:



$$\mathbf{S}_b = \begin{bmatrix} 0 \\ -g\eta\, \partial z_b / \partial x \\ -g\eta\, \partial z_b / \partial y \\ 0 \\ 0 \end{bmatrix}, \quad \mathbf{S}_f = \begin{bmatrix} 0 \\ -fDv \\ fDu \\ 0 \\ 0 \end{bmatrix}, \tag{4}$$

$$\mathbf{S}_{d,h} = \begin{bmatrix} 0 \\ \dfrac{\partial}{\partial x}\left(K_h D \dfrac{\partial u}{\partial x}\right) + \dfrac{\partial}{\partial y}\left(K_h D \dfrac{\partial u}{\partial y}\right) \\ \dfrac{\partial}{\partial x}\left(K_h D \dfrac{\partial v}{\partial x}\right) + \dfrac{\partial}{\partial y}\left(K_h D \dfrac{\partial v}{\partial y}\right) \\ \dfrac{\partial}{\partial x}\left(K_H D \dfrac{\partial T}{\partial x}\right) + \dfrac{\partial}{\partial y}\left(K_H D \dfrac{\partial T}{\partial y}\right) \\ \dfrac{\partial}{\partial x}\left(K_H D \dfrac{\partial S}{\partial x}\right) + \dfrac{\partial}{\partial y}\left(K_H D \dfrac{\partial S}{\partial y}\right) \end{bmatrix}, \quad \mathbf{S}_{d,v} = \begin{bmatrix} 0 \\ \dfrac{\partial}{\partial \sigma}\left(\dfrac{K_v}{D^2}\dfrac{\partial Du}{\partial \sigma}\right) \\ \dfrac{\partial}{\partial \sigma}\left(\dfrac{K_v}{D^2}\dfrac{\partial Dv}{\partial \sigma}\right) \\ \dfrac{\partial}{\partial \sigma}\left(\dfrac{K_V}{D^2}\dfrac{\partial DT}{\partial \sigma}\right) \\ \dfrac{\partial}{\partial \sigma}\left(\dfrac{K_V}{D^2}\dfrac{\partial DS}{\partial \sigma}\right) \end{bmatrix}, \tag{5}$$

$$\mathbf{S}_{baro} = \begin{bmatrix} 0 \\ -\dfrac{gD}{\rho_0}\left[\displaystyle\int_\sigma^0 D\dfrac{\partial \rho}{\partial x}d\sigma - \dfrac{\partial D}{\partial x}\displaystyle\int_\sigma^0 \sigma\dfrac{\partial \rho}{\partial \sigma}d\sigma\right] \\ -\dfrac{gD}{\rho_0}\left[\displaystyle\int_\sigma^0 D\dfrac{\partial \rho}{\partial y}d\sigma - \dfrac{\partial D}{\partial y}\displaystyle\int_\sigma^0 \sigma\dfrac{\partial \rho}{\partial \sigma}d\sigma\right] \\ 0 \\ 0 \end{bmatrix}. \tag{6}$$

where $\eta = D + z_b$ is the surface elevation; $f$ is the Coriolis coefficient; $K_h$ and $K_H$ are the horizontal (eddy) diffusion coefficient using Smagorinsky's method (Smagorinsky, 1963); $K_v$ and $K_V$ are the vertical (eddy) diffusion coefficient calculated by MY-2.5 turbulent closure scheme in a General Ocean Turbulence Model (GOTM, Burchard et al., 1999); $\rho_0$ is

the reference density while the water density $\rho$ is defined as $\rho = \rho(T,S)$, which is a function of $T$ and $S$. In the model, two state equations can be chosen: linear shown in Eq. 7 and full non-linear from Jackett et al. (2006):

$$\rho(T,S) = \rho_0 - \alpha_T(T-T_0) + \beta_S(S-S_0), \tag{7}$$

In Eq. 7, $\alpha_T$ and $\beta_S$ are coefficients. $T_0$ and $S_0$ are reference temperature and salinity respectively. It can be noticed that seven prognostic variables of the above system are $D$ (or $\eta$), $u$, $v$, $\omega$, $\rho$, $T$ and $S$, but there are a total of six equations





(Eqs. 1, 7) to be solved. To this end, the depth-averaged continuity equation is applied to calculate the water depth (or free surface) for the mode nonsplit model, which can be written as:

$$\frac{\partial D}{\partial t} + \frac{\partial DU}{\partial x} + \frac{\partial DV}{\partial y} = 0,$$ (8)

where $U = \int_{-1}^{0} u \, d\sigma$ and $V = \int_{-1}^{0} v \, d\sigma$ are depth-averaged horizontal velocities. The vertical velocity is solved by combining 2D and 3D continuity equations, which can be written as:

$$\frac{\partial \omega}{\partial \sigma} = \frac{\partial (DU - Du)}{\partial x} + \frac{\partial (DV - Dv)}{\partial y},$$ (9)

and then be integrated layer by layer along the vertical direction.

**2.2 Discontinuous Galerkin discretization**

In this section, we describe the numerical implementation of DGCEMS. The DG function spaces are defined in section 2.2.1, followed by the quadrature-free nodal DG discretization (Hesthaven and Warburton, 2007) of the above governing equations.

**2.2.1 Function spaces and notations**

For a 3D computational domain, $\Omega_{3d}$, there is a corresponding projection at the horizontal plane, which is noted as $\Omega_{2d}$, and it can be divided by an unstructured triangular mesh. These triangular elements extend vertically and form triangular prisms, which are computational elements in $\Omega_{3d}$. The DG discretization of this model is based on the linear function space, where in the 2D space is noted as $P_{N_h}$ and in the 3D space is $P_{(N_h, N_v)}$ (here in Figure 1, both $N_h$ and $N_v$ are equal to 1, standing for

the stage of the function of horizontal and vertical space). We set $\phi_{2d} \in P_{N_h}$, $\boldsymbol{\varphi}_{2d} \in P_{N_h}^2$, $\phi_{3d} \in P_{(N_h, N_v)}$ and $\boldsymbol{\varphi}_{3d} \in P_{(N_h, N_v)}^2$ as the test functions in the 2D and 3D function space.

Next, we use the following notations to represent the volume and surface integrals:

$$\langle \cdot \rangle_{\Omega} = \int_{\Omega} (\cdot) \, d\mathbf{x},$$ (10)

$$\langle\langle \cdot \rangle\rangle_{\partial\Omega} = \int_{\partial\Omega} (\cdot) \, ds,$$ (11)

In surface terms, we additionally define the concept of the average, $\{\{\cdot\}\}$, and the jump, $[\![\cdot]\!]$, to simplify the expression of subsequent formulas:

$$\{\{a\}\} = \frac{1}{2}(a^+ + a^-),$$ (12)


$$\{\{\boldsymbol{b}\}\} = \frac{1}{2}\left(\boldsymbol{b}^+ + \boldsymbol{b}^-\right), \tag{13}$$

$$[\![ab]\!] = a^+ \boldsymbol{b}^+ + a^- \boldsymbol{b}^-, \tag{14}$$

$$[\![\boldsymbol{b} \cdot \boldsymbol{n}]\!] = \boldsymbol{b}^+ \cdot \boldsymbol{n}^+ + \boldsymbol{b}^- \cdot \boldsymbol{n}^-, \tag{15}$$

$$[\![\boldsymbol{b}\boldsymbol{n}]\!] = \boldsymbol{b}^+ \boldsymbol{n}^+ + \boldsymbol{b}^- \boldsymbol{n}^-, \tag{16}$$


where $a$ represents a scalar field and $\boldsymbol{b}$ represents a vector field; " $-$ " and " $+$ " are the local side and adjacent side; $\boldsymbol{n} = \left(n_x, n_y, n_\sigma\right)$ is the outward unit normal on the edges. Let $\mathcal{T}$ and $\mathcal{P}$ stand for the partition of the 2D domain $\Omega_{2d}$ and 3D domain $\Omega_{3d}$ respectively. Element interfaces are denoted by $\mathcal{I}_{2d} = \left\{k_i \cap k_j \,\middle|\, k_i, k_j \in \mathcal{T}, i \neq j\right\}$, while $\mathcal{I}_h$ and $\mathcal{I}_v$ are notations of horizontal and vertical interfaces in $\mathcal{P}$.


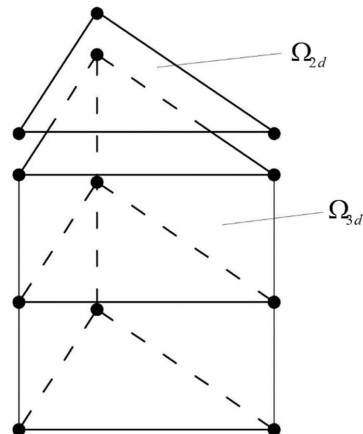

**Figure 1: Computational elements in a vertical two-layer case. Black dots represent the interpolation nodes corresponding to the horizontal one order and vertical one order basis function.**


### 2.2.2 Convection terms and source terms

We multiply Eq. 1 by a test function $\boldsymbol{\varphi} \in \phi_{3d}$ and integrate the face terms by part twice:





$$
\begin{aligned}
&\langle \mathbf{U}_t \cdot \boldsymbol{\varphi} \rangle_{\Omega_d} - \langle \mathbf{F}(\mathbf{U}) \cdot \nabla \boldsymbol{\varphi} \rangle_{\Omega_d} \\
&\quad + \left\langle\!\!\left\langle (\mathbf{E}(\mathbf{U}), \mathbf{G}(\mathbf{U}))^{*,HLLC} \cdot [\![\boldsymbol{\varphi} n_h]\!] \right\rangle\!\!\right\rangle_{\mathcal{I}_v} - \left\langle\!\!\left\langle (\mathbf{E}(\mathbf{U}), \mathbf{G}(\mathbf{U}))^- \cdot [\![\boldsymbol{\varphi} n_h]\!] \right\rangle\!\!\right\rangle_{\mathcal{I}_v} \\
&\quad + \left\langle\!\!\left\langle \mathbf{H}(\mathbf{U})^{*,up} \cdot [\![\boldsymbol{\varphi} n_\sigma]\!] \right\rangle\!\!\right\rangle_{\mathcal{I}_h} - \left\langle\!\!\left\langle \mathbf{H}(\mathbf{U})^- \cdot [\![\boldsymbol{\varphi} n_\sigma]\!] \right\rangle\!\!\right\rangle_{\mathcal{I}_h} \\
&= \langle \mathbf{S}_b \cdot \boldsymbol{\varphi} \rangle_{\Omega_d} + \langle \mathbf{S}_f \cdot \boldsymbol{\varphi} \rangle_{\Omega_d} + \langle \mathbf{S}_{baro} \cdot \boldsymbol{\varphi} \rangle_{\Omega_d} \\
&\quad - \boldsymbol{D}_h(u,v,T,S,\boldsymbol{\varphi}) + \boldsymbol{D}_v(u,v,T,S,\boldsymbol{\varphi}).
\end{aligned}
\tag{17}
$$

Here, $(\quad)^{*,HCCL}$ is calculated by the Harten-Lax-van Leer-Contact (HLLC) method (Toro, 2001); $(\quad)^{*,up}$ means the upwind

numerical flux at the interface; $\boldsymbol{n}_h = (n_x, n_y)$ is the horizontal outward unit normal vector and $n_\sigma$ stands for the vertical

outward unit normal; $\boldsymbol{D}_h$ and $\boldsymbol{D}_v$ are the horizontal and vertical diffusion operators, which will be introduced in the next

section. In computational elements, the bottom and top interfaces are horizontal ($n_x, n_y = 0$) while the side interfaces are

vertical ($n_\sigma = 0$). For convection terms, the process of surface integration is divided into a horizontal part and a vertical part,

which need to use the upwind and the HLLC approximate Riemann solver respectively. This model uses the quadrature-free

nodal DG method to avoid the complex calculation processes of Gaussian integral for nonlinear convection terms. The DG

discretization of the source terms also contains the process of surface integral, but they seem to be omitted in Eq. 17 because

the central numerical flux is used, resulting in the surface integral being zero as a whole.

### 2.2.3 Horizontal and vertical diffusion terms

The discretization of diffusion terms differs from that of convection because they have elliptic operators, which need

additional stabilizations. Thus, the symmetric interior penalty Galerkin (SIPG) method (Epshteyn and Rivière, 2007) is used,

where $\boldsymbol{D}_h$ and $\boldsymbol{D}_v$ read:

$$
\begin{aligned}
\boldsymbol{D}_h(\boldsymbol{u},\boldsymbol{\varphi}) &= -\left\langle \nabla_h \boldsymbol{\varphi} \cdot (D\boldsymbol{v}_h \cdot \nabla_h \boldsymbol{u})^T \right\rangle_{\Omega_d} \\
&\quad + \left\langle\!\!\left\langle \{\!\{ D\boldsymbol{v}_h \cdot \nabla_h \boldsymbol{u} \}\!\} \cdot [\![\boldsymbol{\varphi} n_h]\!] \right\rangle\!\!\right\rangle_{\mathcal{I}_v} + \left\langle\!\!\left\langle \{\!\{ D\boldsymbol{v}_h \cdot \nabla_h \boldsymbol{\varphi} \}\!\} \cdot [\![\boldsymbol{u} n_h]\!] \right\rangle\!\!\right\rangle_{\mathcal{I}_v} \\
&\quad - \left\langle\!\!\left\langle \{\!\{\tau\}\!\} \{\!\{ D\boldsymbol{v}_h \}\!\} [\![\boldsymbol{u} n_h]\!][\![\boldsymbol{\varphi} n_h]\!] \right\rangle\!\!\right\rangle_{\mathcal{I}_v},
\end{aligned}
\tag{18}
$$

$$
\begin{aligned}
\boldsymbol{D}_v(\boldsymbol{u},\boldsymbol{\varphi}) &= -\left\langle \frac{\partial \boldsymbol{\varphi}}{\partial \sigma} \cdot \left( \frac{\boldsymbol{v}_v}{D^2} \cdot \frac{\partial D\boldsymbol{u}}{\partial \sigma} \right) \right\rangle_{\Omega_d} \\
&\quad + \left\langle\!\!\left\langle \left\{\!\!\left\{ \frac{\boldsymbol{v}_v}{D^2} \cdot \frac{\partial D\boldsymbol{u}}{\partial \sigma} \right\}\!\!\right\} \cdot [\![\boldsymbol{\varphi} n_\sigma]\!] \right\rangle\!\!\right\rangle_{\mathcal{I}_h} + \left\langle\!\!\left\langle \left\{\!\!\left\{ \frac{\boldsymbol{v}_v}{D^2} \cdot \frac{\partial \boldsymbol{\varphi}}{\partial \sigma} \right\}\!\!\right\} \cdot [\![ D\boldsymbol{u} n_\sigma ]\!] \right\rangle\!\!\right\rangle_{\mathcal{I}_h} \\
&\quad - \left\langle\!\!\left\langle \{\!\{\tau\}\!\} \left\{\!\!\left\{ \frac{\boldsymbol{v}_v}{D^2} \right\}\!\!\right\} [\![\boldsymbol{u} n_\sigma]\!][\![\boldsymbol{\varphi} n_\sigma]\!] \right\rangle\!\!\right\rangle_{\mathcal{I}_h}.
\end{aligned}
\tag{19}
$$



Here, $\boldsymbol{v}_h = \left(0, K_h, K_h, K_H, K_H\right)^T$ and $\boldsymbol{v}_v = \left(0, K_v, K_v, K_V, K_V\right)^T$; $\tau$ is called the penalty factor. Following the research by

Shahbazi (2005), it can be defined as:

$$\tau = \frac{\gamma}{2} \frac{(N+3)(N+1)}{3L}, \tag{20}$$

in which $N$ represents the order value of the horizontal or vertical basic functions; $\gamma$ is the number of surfaces in an

element, and $L$ is the length scale of the local element in the normal direction of the surface.

### 2.2.4 Primitive continuity and vertical velocity equations

We also multiply Eq. 8 with the test function and integrate the face terms by part twice, the strong form formulation of the

primitive continuity equation is shown in Eq. 20:

$$
\begin{aligned}
\left\langle \frac{\partial D}{\partial t} \boldsymbol{\varphi} \right\rangle_{\Omega_d} &- \left\langle DU \cdot \nabla_h \boldsymbol{\varphi} \right\rangle_{\Omega_d} - \left\langle DV \cdot \nabla_h \boldsymbol{\varphi} \right\rangle_{\Omega_d} \\
&+ \left\langle\!\left\langle (DU)^{*,HLLC} \cdot [\![\boldsymbol{\varphi n}_h]\!] \right\rangle\!\right\rangle_{\mathcal{I}_{2d}} - \left\langle\!\left\langle (DU)^- \cdot [\![\boldsymbol{\varphi n}_h]\!] \right\rangle\!\right\rangle_{\mathcal{I}_{2d}} \\
&+ \left\langle\!\left\langle (DV)^{*,HLLC} \cdot [\![\boldsymbol{\varphi n}_h]\!] \right\rangle\!\right\rangle_{\mathcal{I}_{2d}} - \left\langle\!\left\langle (DV)^- \cdot [\![\boldsymbol{\varphi n}_h]\!] \right\rangle\!\right\rangle_{\mathcal{I}_{2d}} = 0.
\end{aligned}
\tag{21}
$$

Then, expanding the function space of Eq. 21 to the 3D system and subtracting that from the first line of Eq. 17, we have:

$$
\begin{aligned}
\left\langle \frac{\partial \omega}{\partial \sigma} \boldsymbol{\varphi} \right\rangle_{\Omega_d} &= \left\langle (Du - DU) \cdot \nabla_h \boldsymbol{\varphi} \right\rangle_{\Omega_d} + \left\langle (Dv - DV) \cdot \nabla_h \boldsymbol{\varphi} \right\rangle_{\Omega_d} \\
&- \left\langle\!\left\langle (Du - DU)^{*,HLLC} \cdot [\![\boldsymbol{\varphi n}_h]\!] \right\rangle\!\right\rangle_{\mathcal{I}_v} + \left\langle\!\left\langle (Du - DU)^- \cdot [\![\boldsymbol{\varphi n}_h]\!] \right\rangle\!\right\rangle_{\mathcal{I}_v} \\
&- \left\langle\!\left\langle (Dv - DV)^{*,HLLC} \cdot [\![\boldsymbol{\varphi n}_h]\!] \right\rangle\!\right\rangle_{\mathcal{I}_v} + \left\langle\!\left\langle (Dv - DV)^- \cdot [\![\boldsymbol{\varphi n}_h]\!] \right\rangle\!\right\rangle_{\mathcal{I}_v}.
\end{aligned}
\tag{22}
$$

Finally, the vertical integral can achieve vertical velocity from bottom to top.

### 2.3 Slope limiters and wetting/drying treatment

The slope limiters are necessary for the stabilizations of the model. Here we use a 3D anisotropic limiter to control the large

gradient of the horizontal momentum and the tracers. The limited solution at the $j$-th nodal number in the $k$-th triangular

prism element is noted as $\tilde{\mathbf{T}}_{k,j}$, which is defined as:

$$\tilde{\mathbf{T}}_{k,j} = \lambda \mathbf{T}_{k,j} + (1-\lambda)\overline{\mathbf{T}}_k \tag{23}$$





where $\mathbf{T}_{k,j} = \left[ (Du)_{k,j}, (Dv)_{k,j}, (DT)_{k,j}, (DS)_{k,j} \right]^T$ stands for the original solution at the $k$-th triangular prism element in

$\mathcal{P}$; $\overline{\mathbf{T}}_k$ express volume-averaged solutions at the $k$-th triangular prism element, respectively; $\lambda$ is defined by Delandmeter (2017) which ensures the anisotropy of the 3D limiter.

Considering the WD processes in the nearshore regions, the strong discontinuity at the WD fronts also frequently causes

difficulties in the model stability (Le et al., 2020; Medeiros and Hagen, 2013). To avoid the WD problems that may occur in the computational processes, we follow the WD treatment method by Chen et al. (2024), where they combine the vertex-based limiter by Li and Zhang (2017), the positivity-preserving well-balanced limiter proposed by Xing and Zhang (2013) and Eq. 23, to prevent the blow-up of water level and momentum solutions. In this model, the stencil of 2D vertex-based limiter has been adjusted to align with the 3D limiter. This modification was identified through extensive numerical testing

and has been found to enhance computational stability to a certain extent.

**2.4 Time stepping**

In the DG framework, the explicit Runge–Kutta time stepping method is popular. By selecting different numbers of stages, different levels of temporal accuracy can be achieved. Although the explicit scheme has a relatively low computational cost per time step, it imposes strict limitations for time steps. Specifically, for high stiffness terms like the vertical diffusion term,

the required time step should be sufficiently small. Thus, we use the implicit–explicit Runge–Kutta (IMEXRK) scheme. For easy description, the above DG discretization processes are summarized as:

$$\frac{d\boldsymbol{y}}{dt} = \boldsymbol{f}^{\text{EX}}(\boldsymbol{y}) + \boldsymbol{f}^{\text{IM}}(\boldsymbol{y}), \tag{24}$$

where $\boldsymbol{y}$ is noted as degree of freedom on all the elements; $\boldsymbol{f}^{\text{IM}}(\boldsymbol{y})$ stands for the implicit system for the vertical diffusion term while $\boldsymbol{f}^{\text{EX}}(\boldsymbol{y})$ represents explicit system which contains all other terms. The IMEXRK system reads:

$$\boldsymbol{y}^{(1)} = \boldsymbol{y}^{(n)} + \triangle t \boldsymbol{f}^{\text{EX}}\left(\boldsymbol{y}^{(n)}\right), \tag{25}$$

$$\boldsymbol{y}^{(2)} = \boldsymbol{y}^{(n)} + \frac{1}{2}\triangle t\left(\boldsymbol{f}^{\text{EX}}\left(\boldsymbol{y}^{(n)}\right) + \boldsymbol{f}^{\text{EX}}\left(\boldsymbol{y}^{(1)}\right)\right), \tag{26}$$

$$\boldsymbol{y}^{(n+1)} = \boldsymbol{y}^{(2)} + \boldsymbol{f}^{\text{IM}}\left(\boldsymbol{y}^{(n+1)}\right). \tag{27}$$

Here, when the model calculates from step $n$ to $n+1$, it has two middle steps shown in Eq. 25 and 26. Finally, the whole algorithm from the time step $t_n$ to $t_{n+1}$ is shown in Algorithm 1.






---

**Algorithm 1** The calculation process of the model in a computational time step.

---

**First stage:**

1. Calculate the convection term, the horizontal diffusion term, the source terms, and the primitive continuity equation (Eqs. 17, 18 and 21) explicitly according to $D^{(n)}$ (or $\eta^{(n)}$), $Du^{(n)}$, $Dv^{(n)}$, $DT^{(n)}$ and $DS^{(n)}$ at time $t_n$;

2. Achieve the values of $D^{(1)}$ (or $\eta^{(1)}$), $Du^{(1)}$, $Dv^{(1)}$, $DT^{(1)}$ and $DS^{(1)}$ at the first middle step (Eq. 25);

3. Apply slope limiters to $Du^{(1)}$, $Dv^{(1)}$, $DT^{(1)}$ and $DS^{(1)}$ (Eq. 23);

4. Calculate vertical-averaged horizontal velocities by integration and update the WD statement, the density $\rho^{(n)}$ and vertical velocity $\omega^{(n)}$ (Eq. 7 and 9).

**Second stage:**

5. Calculate the convection term, the horizontal diffusion term, the source terms, and the primitive continuity equation (Eqs. 17, 18 and 21) explicitly according to $D^{(1)}$ (or $\eta^{(1)}$), $Du^{(1)}$, $Dv^{(1)}$, $DT^{(1)}$ and $DS^{(1)}$;

6. Achieve the values of $D^{(n+1)}$ (or $\eta^{(n+1)}$), $Du^{(2)}$, $Dv^{(2)}$, $DT^{(2)}$ and $DS^{(2)}$ at the second middle step using the values at time $t_n$ and the first middle step (Eq. 26);

7. Apply slope limiters to $Du^{(2)}$, $Dv^{(2)}$, $DT^{(2)}$ and $DS^{(2)}$ (Eq. 23);

**Final stage:**

8. Calculate the vertical viscosity and vertical diffusion implicitly;

9. Achieve the values of $Du^{(n+1)}$, $Dv^{(n+1)}$, $DT^{(n+1)}$ and $DS^{(n+1)}$ at time $t_{n+1}$ (Eq.27);

10. Calculate vertical-averaged horizontal velocities by integration and update the WD statement, the density $\rho^{(n+1)}$ and vertical velocity $\omega^{(n+1)}$ (Eq. 7 and 9).

---

## 3 Tests and analysis

In this section, we first validate the baroclinic solver by a manufactured solution test and then verify the diffusion terms using a standard lock exchange test. An ideal river plume test case and a semi-closed estuary case of salinity intrusion are also applied to examine the performance of DGCEMS.

### 3.1 Baroclinic manufactured solution test

The computational domain is a rectangular box of length $L_x = 15 \text{ km}$, width $L_y = 10 \text{ km}$, and depth $D = 40 \text{ m}$. At the initial time step, we define the velocity field and the tracer field as follows:



$$u = \frac{1}{2}\sin\left(\frac{2\pi x}{L_x}\right)\cos(3\sigma), \tag{28}$$

$$v = \frac{1}{3}\cos\left(\frac{\pi y}{L_y}\right)\sin\left(\frac{\sigma}{2}\right), \tag{29}$$

$$T = 15 + 15\sin\left(\frac{\pi x}{L_x}\right)\sin\left(\frac{\pi y}{L_y}\right)\cos(\sigma), \tag{30}$$

where the above three expressions are also represented using $\sigma$ coordinate, ranging from -1 to 0, at the vertical direction. In this case, the Coriolis parameter $f = 0.0001$, while the bottom friction term, the viscosity and diffusion terms are omitted.

The linear state equation is applied with $\rho_0 = 1000 \ \mathrm{kgm}^{-3}$, $\alpha_T = 0.2 \ \mathrm{kgm}^{-3}\,^{\circ}\mathrm{C}^{-1}$ and $T_0 = 5.0 \ ^{\circ}\mathrm{C}$. The value of $\beta_s$ is not defined because the salinity in the whole field is a constant. We set 10 layers at vertical directions with the horizontal mesh resolutions of 2500 m, 1250 m, 625 m, 312.5 m, and 156.25 m, respectively, where the grid shapes are isosceles right-angled triangle. In the computational domain, all the boundary edges are closed and no external force needs to be applied, which cause the system a time-dependent problem. To achieve a steady state solution, we derived the analytical functions of the advection term, the baroclinic term, and the Coriolis acceleration term and then added them to the source term to balance the initial velocity and tracer field. Details can be found in Appendix A.

For the five grid resolutions mentioned above, the model was run for 100 steps with time steps of 4.0 s, 2.0 s, 1.0 s, 0.5 s, and 0.25 s, respectively. The $L_2$ errors of the water elevation, horizontal velocity, vertical velocity, and temperature field across the entire computational domain were calculated, as shown in Figure 2. The red lines shown in the figure represents the best-fit line obtained using the least-squares method, with its slope indicating the order of convergence. For varying grid resolutions, the water elevation, horizontal velocity, and temperature exhibit nearly second-order convergence. However, the vertical velocity fails to achieve second-order convergence, which is attributed to the use of the vertically averaged continuity equation in the vertical velocity computation. The numerical solution of this equation inherently contains errors that accumulate throughout the calculation process, preventing the vertical velocity from reaching optimal second-order convergence. Overall, the results from this artificial analytical solution test are consistent with those of Kärnä et al. (2018), confirming that the discretization of the advection term, baroclinic term, and Coriolis term in $\sigma$ coordinates is accurate at both the coding and algorithmic levels.



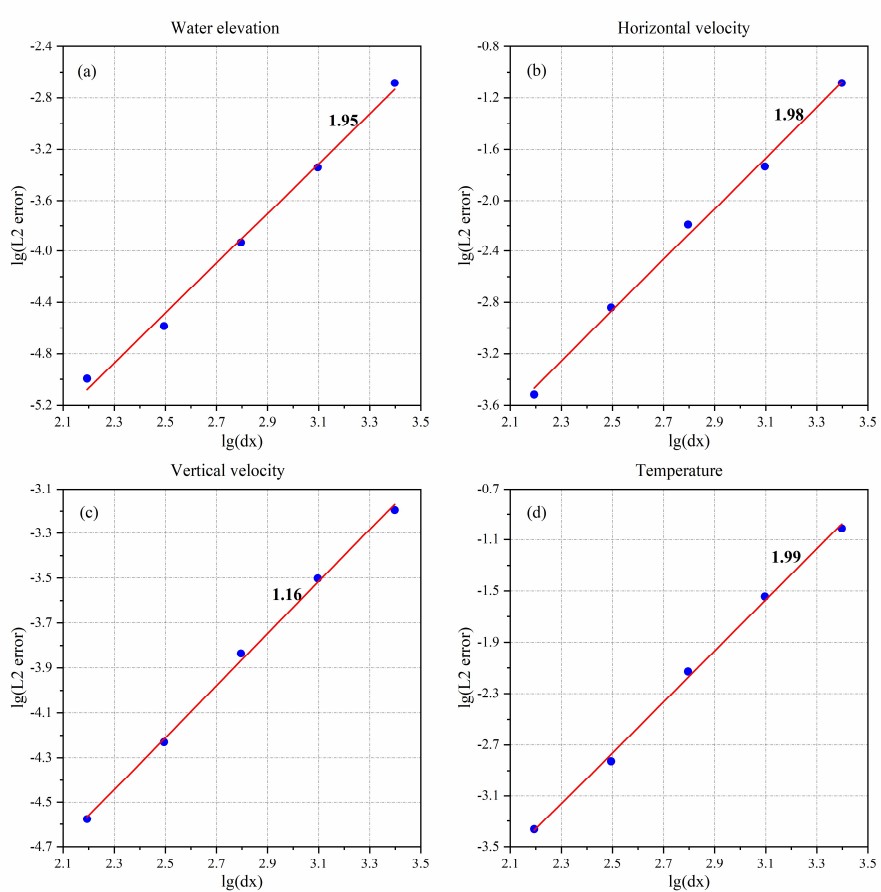

**Figure 2: Convergence of the $L_2$ error of (a) surface elevation, (b) horizontal velocity, (c) vertical velocity, and (d) temperature field in the baroclinic manufactured solution test case.**

**3.2 Lock exchange**

In this subsection, we validate the baroclinic term and discuss the spurious mixing caused by horizontal diffusion with a standard lock exchange test (Ilıcak et al., 2012; Petersen et al., 2015; Kärnä et al., 2018). The computational domain is a rectangular channel which is 64-km-long, 1-km-wide, and 20-m-depth. Each element in the domain is $\Delta x = 500$ m of the triangle edge length and 1m in the vertical direction (total of 20 layers). The initial salinity is set to a constant value of 35psu

in the whole domain and the initial temperature is calculated by:

$$T(x, y, \sigma) = 5 \, {}^{\circ}\text{C} \quad \text{where } 0 \leq x < 32 \text{ km,} \tag{31}$$



$$T\left(x,y,\sigma\right)=35\ ^{\circ}\text{C}\ \text{where}\ 32\leq x\leq 64\ \text{km.} \tag{32}$$

We use Eq. 7 to update the density, where $\alpha_T = 0.2$, $\beta_S = 0$, $T_0 = 5\ ^{\circ}\text{C}$ and $S_0 = 35\ \text{psu}$. At the location of $x = 32\ \text{km}$, the initial density difference $\triangle\rho = 6.0\ \text{kgm}^{-3}$. In the model, only the convective term controls the tracer concentration and

the bottom friction is omitted. The vertical viscosity $K_v$ is $10^{-4}\ \text{m}^2\text{s}^{-1}$ and the horizontal viscosity $K_h$ is varied with the value of 1, 10, 200 $\text{m}^2\text{s}^{-1}$ so that the grid Reynolds numbers $\text{Re} = U_{grid}\Delta x/K_h = 250.0,\ 25.0,\ \text{and}\ 1.25$, where $U_{grid} = 0.5\ \text{ms}^{-1}$ represents the characteristic velocity scale.

Figure 3a shows the initial density field, and Figure 3b, 3c, 3d are the density field at the time of 17 hours where the Re are 250.0, 25.0, and 1.25, respectively. It seems that as the Re decreases, the density front becomes smoother, indicating that the

increase in background viscosity reduces overall mixing within the system. To quantify the role of spurious mixing, we also use a reference potential energy (RPE), which can be defined as:

$$\text{RPE} = g\int\rho^*\left(z+D\right)\text{d}z, \tag{33}$$

where $\rho^*$ is rearranged in descending order based on actual density values $\rho$ from bottom to top. Under this computational approach, RPE represents the potential energy within the system that cannot be converted into kinetic energy. When spurious

mixing is present, the value of RPE in the system increases. At any given time, $t$, the dimensionless relative reference potential energy ($\overline{\text{RPE}}(t)$) is defined as follows:

$$\overline{\text{RPE}}(t) = \frac{\text{RPE}(t) - \text{RPE}(0)}{\text{RPE}(0)}. \tag{34}$$

The calculation results of $\overline{\text{RPE}}(t)$ for the three Reynolds numbers are shown in Figure 4a. At the 17th hour, the values of $\overline{\text{RPE}}(17)$ are 3.06, 2.26, and $0.66\times10^{-5}$ respectively, which is agreed with previous simulating results (Ilıcak et al., 2012;

Kärnä et al., 2018). Additionally, the locations of the density fronts simulated under three different grid Reynolds number conditions were captured and compared with the theoretical maximum propagation distance, $L$, where $L = t/2\sqrt{gD\Delta\rho/\rho_0}$ The spurious mixing resulted in a loss of potential energy throughout the system, leading to discrepancies between the model simulations and the theoretical values (Figure 4b). Overall, the error remains within 4.7%, which is consistent with Kärnä et al. (2018).



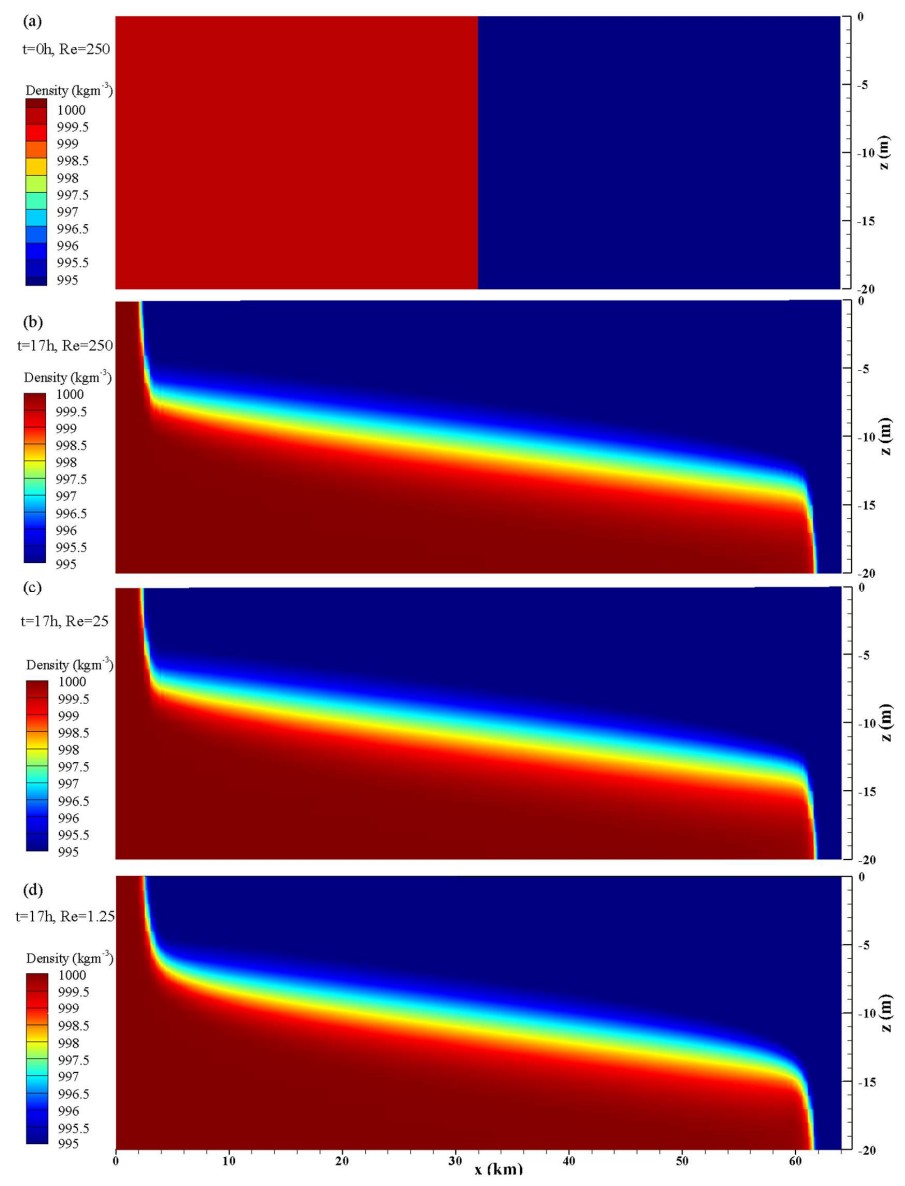

**Figure 3: The density in the lock exchange test under three different grid Reynolds numbers.**




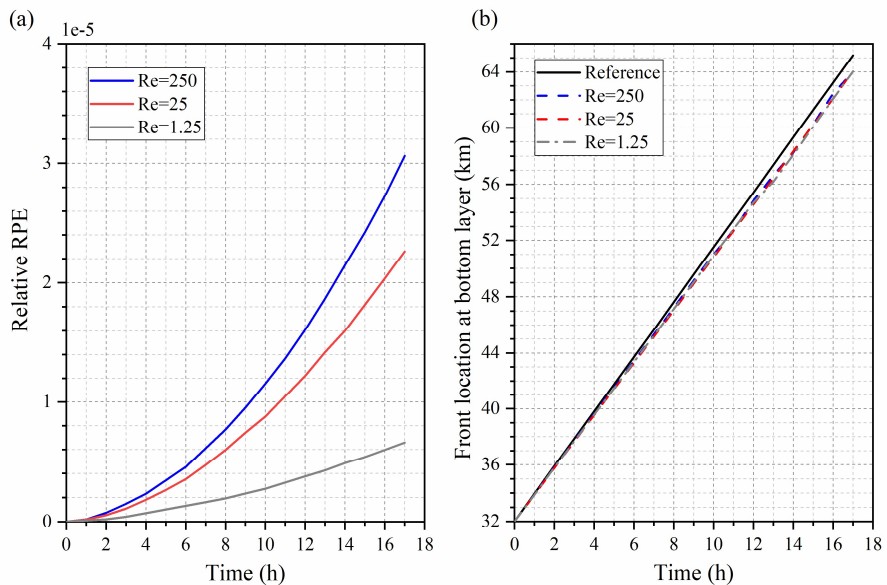

**Figure 4: Simulating results of (a) relative RPE and (b) the density front location at bottom layer of the lock exchange test with different Re.**

### 3.3 An ideal river plume

We continue to experiment with an ideal river plume (Wu 2023) to analyse the model's performance. Saltwater can be replaced by various contaminants that affect water quality. Therefore, investigating surface plume dynamics not only validates the model's accuracy but also highlights its applicability to environmental modelling and pollutant transport simulations.

### 3.3.1 Settings

Figure 5 shows the topography of the estuary region, where there is a 7.5-km-long, 3.0-km-wide, and 15-m-depth river that connects to the sea. The initial computational mesh domain is shown in Figure 6. A freshwater runoff with a rate of 1500 $m^3s^{-1}$ and a salinity of zero flows into a shelf with its bottom slope of 0.0007, where the salinity of ocean water is 32 psu. We first use a grid resolution of 1500 m to partition the computational domain, which means that there are only two rows of grids along the width of the river channel. In this test, the Coriolis parameter is calculated by the Latitude of $45°\,N$, which is about 0.0001, the bottom friction coefficient is calculated by MY-2.5 turbulent closure scheme with 0.2mm of the bottom roughness parameter, and the eddy diffusivity coefficient is set to zero. Besides, other external forces are not considered in



this test. The open boundary is set to radiative according to Wu (2023), so that the boundary will have less influence on the currents in the computational domain. We also use Eq. 7 to update the density, where $\alpha_T = 0$, $\beta_S = 0.78$, $T_0 = 20^\circ C$ and

$S_0 = 32$ psu .

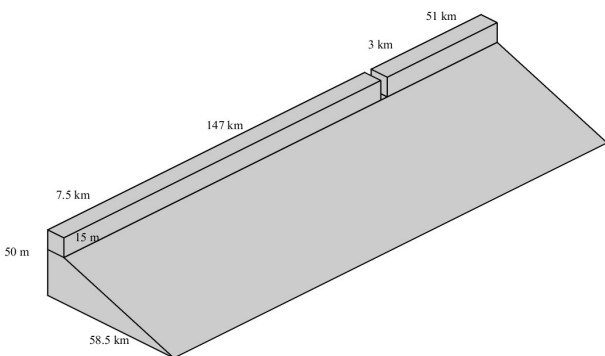

**Figure 5: The topography of the ideal river plume experiment.**

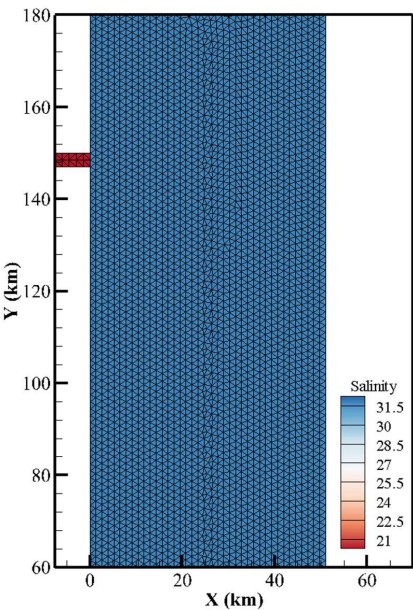

**Fig. 6 The computational mesh domain and the initial salinity field. The salinity is 0 psu at x < 0 km and 32psu at x≥0 km.**


### 3.3.2 Analysis of plume pattern

Due to the lack of analytical solutions in the test, we will illustrate the accuracy of the simulation results from the perspective of grid convergence. For numerical methods, we prefer them to be less affected by grid resolution. Here, the mesh resolution is refined by a factor of 2 and 4 in both $x$ and $y$ directions and the simulated results at 48 hours are shown in Figure 7. Considering the impact of differences in grid resolution on images, the simulation results of the three types of grids were processed in the same way to minimize the impact of post-processing as much as possible. Based on the simulated plume pattern, as the computational grid resolution increases, the zig-zag phenomenon at the plume front gradually diminishes, resulting in a smoother edge and a more pronounced bulge shape. Simultaneously, there are noticeable differences in the extent of freshwater transport along the coast; with higher grid resolution, the transport distance increases. This may be attributed to the finer grid's ability to more accurately capture buoyancy fronts, thereby enhancing the coastal extension of freshwater. Additionally, the finer resolution reduces numerical dissipation, contributing to the observed differences in transport distance.

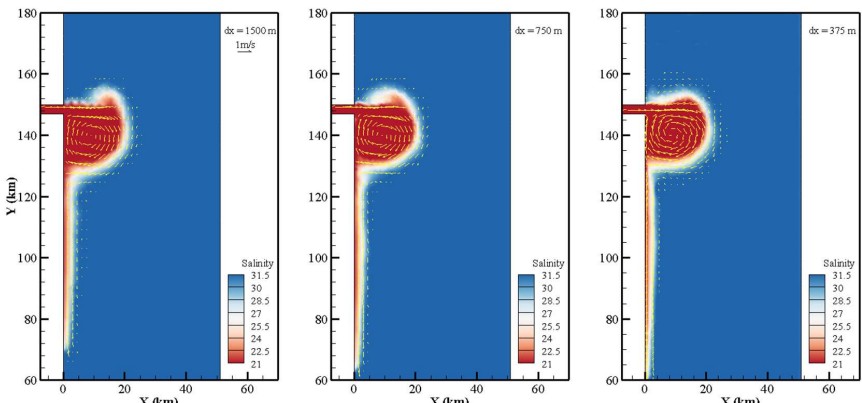

**Fig. 7 Simulated surface river plume and surface current velocity at 48 hours. The grid resolution is refined from 1500m (left panel) to 375 m (right panel).**

### 3.3.3 Quantitative analysis

To further demonstrate the performance of the model, a series of quantitative methods will be employed to analyse the numerical simulation results of freshwater content ($Fc$ in short), plume characteristics in the near- and mid-field regions, as well as coastal plume transport in the far-field region under different computational grid resolutions.

Following the concept of isohaline coordinate by MacCready et al. (2002), the calculation method of freshwater volume can be written as:




$$V\left(S^*\right)=\int_{S<S^*}\left(S_0-S\right)/S_0 dV, \tag{35}$$

where $S^*$ represents each salinity level. Then, we have:

$$Fc\left(S\right)=\frac{dV\left(S\right)}{dS}. \tag{36}$$

It can be shown in Figure 8 that under the three grid resolutions, the freshwater content exhibits a similar trend: it increases initially and then decreases as salinity rises, with the peak freshwater content occurring at a salinity of 26 psu. This behaviour can be explained by the relationship between salinity and the distance from the river mouth. Lower salinity values indicate proximity to the river mouth, where the number of grids is relatively small, resulting in lower freshwater content in these regions. Conversely, higher salinity values correspond to regions farther from the river mouth, where more grids are

present, but the freshwater content per cell is low, leading to lower overall freshwater content in high-salinity areas. These results are consistent with Wu's research.

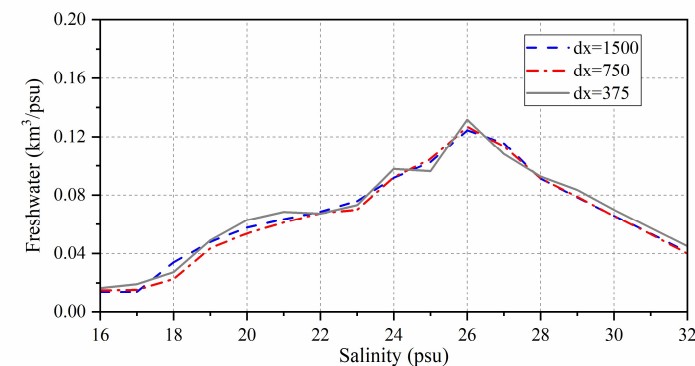

**Fig. 8 Freshwater content at per salinity class at 48 hours' results with different grid resolution.**

Secondly, at $y$ = 140 km, approximately at the location of the plume's rotational center, a cross-sectional profile was extracted to analyse the effect of grid resolution on the formation of the bulge in the near- and mid-field regions (see Figure 9). When the grid resolution is 1500 m, the salinity at the center of the bulge is relatively low, around 17.5 psu. As the grid resolution increases, the asymmetry of surface salinity from the center of the bulge toward the edge of the inertial radius becomes more pronounced. Using the 375 m grid resolution as the reference, when the grid resolution is doubled or

quadrupled, the salinity error relative to the reference case fluctuates within 10%.

Finally, at $y$ = 100 km, the relative freshwater transport carried by the coastal buoyancy current induced by the plume is calculated to assess the impact of grid resolution on the simulation results in the far-field region. The relative freshwater transport is determined by the ratio of the freshwater content at this location to that at the river mouth. At $y$ = 100 km, the

simulated surface coastal current velocity is consistently 0.30 ms⁻¹ across all grid resolutions, indicating minimal influence of

grid resolution on current velocity. The percentage of freshwater transport at this location is 59.4%, 57.5%, and 60.6% for

grid resolutions of 1500 m, 750 m and 375 m, respectively.

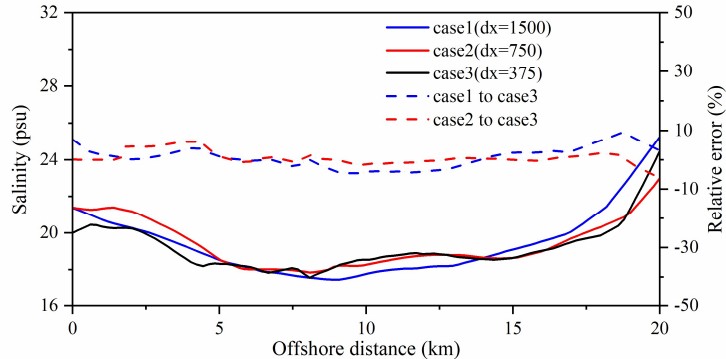

**Fig. 9 Surface salinity at the profile of *y* = 140 km under three grid resolutions. Blue and red dashed lines are salinity relative errors with grid size 1500 m and 750 m respectively compared to that of 375 m.**

### 355   3.4 Saltwater enters an idealized semi-enclosed estuary

In addition to estuarine freshwater plumes, another phenomenon related to saltwater and freshwater interactions is saltwater intrusion into estuaries under tidal forcing. In this section, a semi-enclosed idealized river channel is configured as shown in Figure 10. The channel is 15 km long, 3 km wide, and has a depth of 10.2 m, with an initial salinity of 0. The channel features a riverbank slope of 0.033 and a floodplain slope of 0.0007. The right boundary is open to the sea, with a salinity of

35psu, where a semi-diurnal tide with a 1 m amplitude is applied. The water temperature is uniformly set to 20°C. Given that existing 3D DG baroclinic models lack empirical validation for handling WD processes, this test examines the model's applicability and reliability in solving the salinity transport equation in the presence of WD boundaries. The computational domain features a grid resolution ranging from 200m to 800m, with the vertical dimension divided into 10 layers. The model operates with a time step of 0.2 s. The linear state equation is again applied, with $\alpha_T = 0$, $\beta_S = 0.714$, $T_0 = 20°$C and

$S_0 = 35$ psu, to update the density. The horizontal and vertical viscosities are set to 0.01 m²s⁻¹, the bottom friction coefficient is set to 0.005, and the threshold water depth is set to 0.05 m. Limiters mentioned in section 2.3 are all applied to maintain robustness.





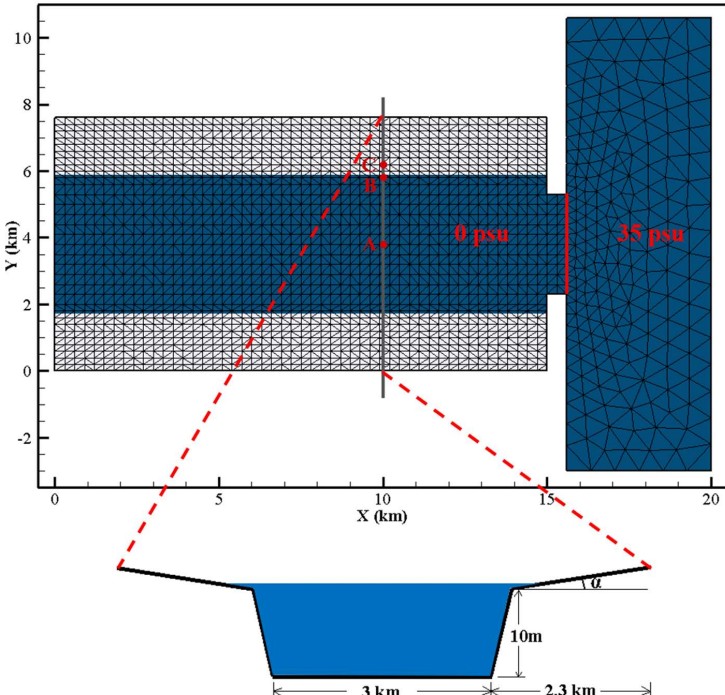

**Fig. 10 The calculation domain and its grid division of the semi-closed estuary. The minimum grid resolution is 200 m. The river**
**channel cross-section is symmetric about the channel centerline and uniform along the channel direction. The initial water depth is**
**10.2 m and $\alpha = 0.0007$. Three characteristic points are marked: the channel center (A), the WD boundary (B), and the initially dry**
**location (C). The initial salinity is set to 0 on the left side of the red solid line and 35 psu on the right side.**

During the flood tide, seawater from the open boundary flows into the semi-closed channel and overtops the floodplain
(Figure 11). Three representative points were selected to analyse temporal variations in tidal water elevation and velocities:
site A, located at the center of the channel; site B, near the WD interface on the top of the riverbank slope at the initial time;
and site C, situated in the initially dry intertidal zone (Figure 10). At sites A and B, the surface water elevations exhibit
nearly identical variations over a lunar day. However, at site C, water levels show periodic fluctuations only during the 8
hours corresponding to the high-water time, remaining at zero for the rest of the time (Figure 12a). For site A, the mean
velocity in the along-channel direction exhibits a well-defined periodic variation, while the mean velocity in the cross-
channel direction remains close to zero, indicating that the velocity field within the channel is minimally affected by the WD
boundary treatment. For site B, the mean velocity in the along-channel direction is generally similar to that of site A.
However, during the two flood tide phases, the velocity is slightly reduced due to the influence of the wetting and drying



boundary treatment. In contrast, the mean velocity in the cross-channel direction shows periodic variation consistent with the

tidal cycle. At site C, velocity is only observed during high water levels, with both the magnitude and direction of the velocity varying during the 8-hour window of tidal inundation (Figures 12b, 12c). Besides, the salinity variations along the channel centerline are shown in Figure 13. As seawater intrudes from the estuarine mouth, the salinity values within the channel progressively increase in the negative $x$-direction. Vertically, the salinity distribution exhibits a characteristic pattern of lower salinity near the surface and higher salinity near the bottom. Although this numerical experiment may not provide

highly quantitative conclusions, the model's simulation results of the saltwater intrusion process involving WD interfaces are consistent with physical expectation.

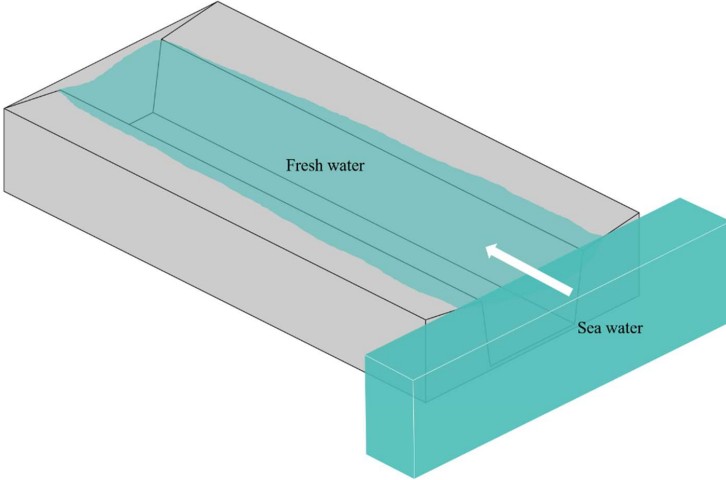

**Fig. 11 A 3D view of a flooding time.**

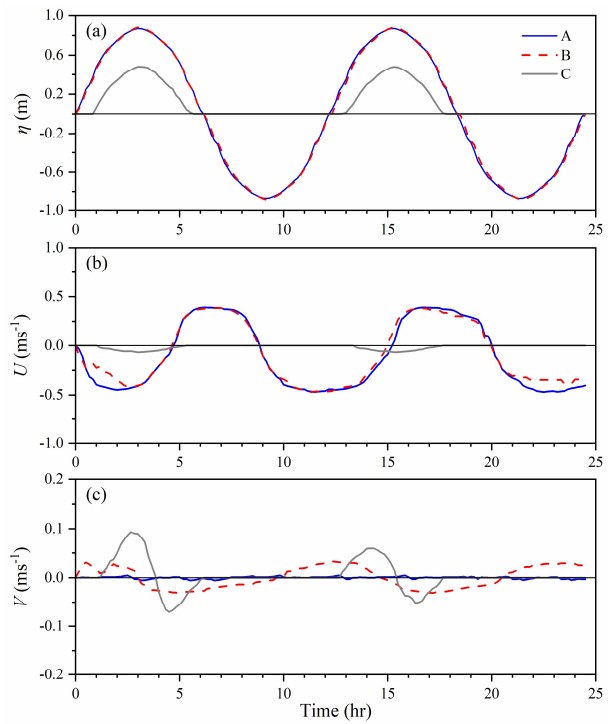

**Fig. 12 Time series of surface water elevation (a), depth-averaged velocity along the channel (b) and cross the channel (c) at points A, B, and C.**

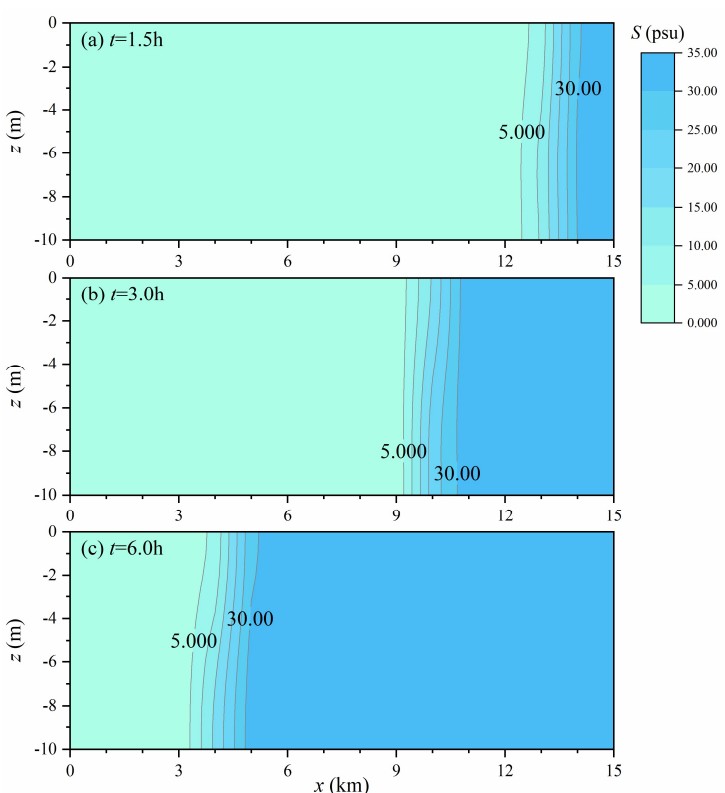

**Fig. 13 Salinity distribution along the channel centerline cross-section at 1.5 h, 3 h, and 6 h.**

**4 Conclusions**

This study presents the development of a novel three-dimensional discontinuous Galerkin coastal and estuarine modelling system, DGCEMS. The model distinguishes itself from existing 3D DG-based ocean models by employing σ-coordinates, a non-split mode framework and a quadrature-free nodal discontinuous Galerkin method. A semi-implicit Runge–Kutta scheme is applied to ensure a second-order accuracy in both space and time. Numerical tests show that the simulation results of the model have a second-order convergence of surface water elevation, horizontal velocity and tracer field, while the convergence of vertical velocity field is about one order. The spurious mixing is also well-controlled and comparable to the existing 3D DG coastal ocean model, Thetis. The new model demonstrates low sensitivity to changes in grid resolution when





simulating surface river plumes, consistently producing comparable results even with relatively coarse grids, and it has the capability to simulate salt-freshwater interactions in the presence of wetting and drying boundaries.

**Acknowledgments**

The authors gratefully acknowledge the financial support by the Joint Funds of the National Natural Science Foundation of China (Grant no. U1906231).

**Code availability**

The static version of the DGCEMS(v1.0.0) source code is available at https://doi.org/10.5281/zenodo.14803723. The code
of DGCEMS is publicly available on the author's Github via https://github.com/ZerengChen/DGCEMS.

**Data availability**

No external data were used in this paper.

**Author contributions**

Chen. Z.R.: Conceptualization, Methodology, Software, Validation, Writing - Original Draft, Visualization;
Zhang. Q.H.: Conceptualization, Writing - Review & Editing, Supervision, Project administration, Funding acquisition;
Ran. G.Q.: Methodology, Software, Supervision;
Nie. Y.: Validation, Formal analysis, Visualization.

**Competing interests**

The contact author has declared that neither they nor their co-authors have any competing interests.



## Appendix A: Terms for the baroclinic manufactured solution test

Using Eq. (28)-(30), the steady-state solution for other fields and terms are:

$$\eta = 0, \tag{A1}$$

$$U = \frac{1}{6}\sin(3)\sin\left(\frac{2\pi x}{L_x}\right), \tag{A2}$$

$$V = \frac{-4}{3}\sin^2\left(\frac{1}{4}\right)\cos\left(\frac{\pi y}{L_y}\right), \tag{A3}$$

$$\omega = -\frac{D\pi}{3L_xL_y}\left(L_y\cos\left(\frac{2\pi x}{L_x}\right)(\sin(3)+\sin(3\sigma)) + 2L_x\left(-\cos\left(\frac{1}{2}\right)+\cos\left(\frac{\sigma}{2}\right)\right)\sin\left(\frac{\pi y}{L_y}\right)\right), \tag{A4}$$

$$\frac{\partial DU}{\partial x} + \frac{\partial DV}{\partial y} = \frac{D\pi}{3L_xL_y}\left(L_y\sin(3)\cos\left(\frac{2\pi x}{L_x}\right) + 4\sin^2\left(\frac{1}{4}\right)L_x\sin\left(\frac{\pi y}{L_y}\right)\right), \tag{A5}$$

$$\frac{\partial Duu}{\partial x} = \frac{D\pi}{2L_x}\cos^2(3\sigma)\sin\left(\frac{4\pi x}{L_x}\right), \tag{A6}$$

$$\frac{\partial Duv}{\partial y} = -\frac{D\pi}{6L_y}\cos(3\sigma)\sin\left(\frac{\sigma}{2}\right)\sin\left(\frac{2\pi x}{L_x}\right)\sin\left(\frac{\pi y}{L_y}\right), \tag{A7}$$

$$\frac{\partial Duv}{\partial x} = \frac{D\pi}{3L_x}\cos(3\sigma)\sin\left(\frac{\sigma}{2}\right)\cos\left(\frac{2\pi x}{L_x}\right)\cos\left(\frac{\pi y}{L_y}\right), \tag{A8}$$

$$\frac{\partial Dvv}{\partial y} = -\frac{D\pi}{9L_y}\sin^2\left(\frac{\sigma}{2}\right)\sin\left(\frac{2\pi y}{L_y}\right), \tag{A9}$$

$$\frac{\partial u\omega}{\partial \sigma} = \frac{D\pi\sin(3\sigma)\sin(2\pi x/L_x)}{2L_xL_y}$$
$$\cdot\left(L_y\cos\left(\frac{2\pi x}{L_x}\right)(\sin(3)+\sin(3\sigma)) + 2L_x\left(-\cos\left(\frac{1}{2}\right)+\cos\left(\frac{\sigma}{2}\right)\right)\sin\left(\frac{\pi y}{L_y}\right)\right)$$
$$-\frac{D\pi\cos(3\sigma)\sin(2\pi x/L_x)}{6L_xL_y}\left(3L_y\cos(3\sigma)\cos\left(\frac{2\pi x}{L_x}\right) - L_x\sin\left(\frac{\sigma}{2}\right)\sin\left(\frac{\pi y}{L_y}\right)\right), \tag{A10}$$





$$\frac{\partial v\omega}{\partial \sigma} = -\frac{D\pi \cos(\sigma/2)\cos(\pi y/L_y)}{18 L_x L_y}$$
$$\cdot \left( L_y \cos\left(\frac{2\pi x}{L_x}\right)(\sin(3)+\sin(3\sigma)) + 2L_x\left(-\cos\left(\frac{1}{2}\right)+\cos\left(\frac{\sigma}{2}\right)\right)\sin\left(\frac{\pi y}{L_y}\right)\right)$$
$$-\frac{D\pi \cos(\pi y/L_y)\sin(\sigma/2)}{9 L_x L_y}\left(3L_y \cos(3\sigma)\cos\left(\frac{2\pi x}{L_x}\right) - L_x \sin\left(\frac{\sigma}{2}\right)\sin\left(\frac{\pi y}{L_y}\right)\right), \tag{A11}$$

$$\frac{\partial DuT}{\partial x} = \frac{15 D\pi \cos(3\sigma)}{4 L_x}\left(4\cos\left(\frac{2\pi x}{L_x}\right)+\cos(\sigma)\left(-\sin\left(\frac{\pi x}{L_x}\right)+3\sin\left(\frac{3\pi x}{L_x}\right)\right)\sin\left(\frac{\pi y}{L_y}\right)\right), \tag{A12}$$

$$\frac{\partial DvT}{\partial y} = \frac{5 D\pi \sin(\sigma/2)}{L_y}\left(\cos(\sigma)\cos\left(\frac{2\pi y}{L_y}\right)\sin\left(\frac{\pi x}{L_x}\right)-\sin\left(\frac{\pi y}{L_y}\right)\right), \tag{A13}$$

$$\frac{\partial \omega T}{\partial \sigma} = \frac{5 D\pi \sin(\sigma)\sin(\pi x/L_x)\sin(\pi y/L_y)}{L_x L_y}$$
$$\cdot \left(L_y \cos\left(\frac{2\pi x}{L_x}\right)(\sin(3)+\sin(3\sigma))+2L_x\left(-\cos\left(\frac{1}{2}\right)+\cos\left(\frac{\sigma}{2}\right)\right)\sin\left(\frac{\pi y}{L_y}\right)\right)$$
$$-\frac{5D\pi}{L_x L_y}\left(3L_y \cos(3\sigma)\cos\left(\frac{2\pi x}{L_x}\right)-L_x \sin\left(\frac{\sigma}{2}\right)\sin\left(\frac{\pi y}{L_y}\right)\right)\left(1+\cos(\sigma)\sin\left(\frac{\pi x}{L_x}\right)\sin\left(\frac{\pi y}{L_y}\right)\right), \tag{A14}$$

$$S_{baro,x} = -\frac{15\alpha_T D^2 g\pi}{L_x \rho_0}\cos\left(\frac{\pi x}{L_x}\right)\sin(\sigma)\sin\left(\frac{\pi y}{L_y}\right), \tag{A15}$$

$$S_{baro,y} = -\frac{15\alpha_T D^2 g\pi}{L_y \rho_0}\cos\left(\frac{\pi y}{L_y}\right)\sin(\sigma)\sin\left(\frac{\pi x}{L_x}\right), \tag{A16}$$

$$S_{f,x} = -\frac{fD}{3}\cos\left(\frac{2\pi y}{L_y}\right)\sin\left(\frac{\sigma}{2}\right), \tag{A17}$$

$$S_{f,y} = \frac{fD}{2}\sin\left(\frac{2\pi x}{L_x}\right)\cos(3\sigma). \tag{A18}$$

The above terms are added to the right-hand side of Eq. (3) like source terms to balance the initial fields.



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
