# Peer review of "The discontinuous Galerkin coastal and estuarine modelling system (DGCEMS v1.0.0): a three-dimensional, mode-nonsplit, implicit-explicit Runge-Kutta hydrostatic model"

_Geoscientific Model Development, 2024_

## Author Comment (AC3)

Thank you for your suggestion and understanding. We will include a brief theoretical analysis of the second-order temporal accuracy in the appendix.

The IMEX Runge–Kutta (IMEXRK) scheme used in this work are shown in Eq.(24)-(27):

$$\frac{d\mathbf{y}}{dt} = \mathbf{f}^{EX}(\mathbf{y}) + \mathbf{f}^{IM}(\mathbf{y}), \tag{24}$$

$$\mathbf{y}^{(1)} = \mathbf{y}^{(n)} + \Delta t \mathbf{f}^{EX}(\mathbf{y}^{(n)}), \tag{25}$$

$$\mathbf{y}^{(2)} = \mathbf{y}^{(n)} + \frac{1}{2}\Delta t\left(\mathbf{f}^{EX}(\mathbf{y}^{(n)}) + \mathbf{f}^{EX}(\mathbf{y}^{(1)})\right), \tag{26}$$

$$\mathbf{y}^{(n+1)} = \mathbf{y}^{(2)} + \Delta t \mathbf{f}^{IM}(\mathbf{y}^{(n+1)}). \tag{27}$$

For explicit term, we first expand $\mathbf{f}^{EX}(\mathbf{y}^{(1)})$:

$$\mathbf{f}^{EX}(\mathbf{y}^{(1)}) = \mathbf{f}^{EX}(\mathbf{y}^{(n)} + \Delta t \mathbf{f}^{EX}(\mathbf{y}^{(n)})) = \mathbf{f}^{EX}(\mathbf{y}^{(n)}) + \Delta t \mathbf{f}^{(EX)'}(\mathbf{y}^{(n)})\mathbf{f}^{EX}(\mathbf{y}^{(n)}) + O(\Delta t^2). \tag{B1}$$

Substituting into (26), we obtain:

$$\mathbf{y}^{(2)} = \mathbf{y}^{(n)} + \Delta t \mathbf{f}^{EX}(\mathbf{y}^{(n)}) + \frac{\Delta t^2}{2}\mathbf{f}^{(EX)'}(\mathbf{y}^{(n)})\mathbf{f}^{EX}(\mathbf{y}^{(n)}) + O(\Delta t^3). \tag{B2}$$

The implicit term can be written via a fixed-point expansion:

$$\mathbf{y}^{(n+1)} = \mathbf{y}^{(2)} + \Delta t \mathbf{f}^{IM}(\mathbf{y}^{(2)}) + O(\Delta t^2). \tag{B3}$$

Substituting (B2) into (B3), we obtain:

$$\mathbf{y}^{(n+1)} = \mathbf{y}^{(n)} + \Delta t \mathbf{f}^{EX}(\mathbf{y}^{(n)}) + \frac{\Delta t^2}{2}\mathbf{f}^{(EX)'}(\mathbf{y}^{(n)})\mathbf{f}^{EX}(\mathbf{y}^{(n)}) + O(\Delta t^3)$$
$$+ \Delta t \mathbf{f}^{IM}(\mathbf{y}^{(2)}) + O(\Delta t^2). \tag{B4}$$

Then, we expand $\mathbf{f}^{IM}(\mathbf{y}^{(2)})$ and it:

$$\mathbf{f}^{IM}(\mathbf{y}^{(2)}) = \mathbf{f}^{IM}(\mathbf{y}^{(n)}) + \frac{\Delta t}{2}\mathbf{f}^{(IM)'}(\mathbf{y}^{(n)})\mathbf{f}^{IM}(\mathbf{y}^{(n)}) + O(\Delta t^2). \tag{B5}$$

We can find that $\mathbf{f}^{IM}(\mathbf{y}^{(2)})$ is first-order and $\Delta t \mathbf{f}^{IM}(\mathbf{y}^{(2)})$ is second-order. Substituting (B5) into (B4), we obtain:

$$\mathbf{y}^{(n+1)} = \mathbf{y}^{(n)} + \Delta t \mathbf{f}^{EX}(\mathbf{y}^{(n)}) + \frac{\Delta t^2}{2}\mathbf{f}^{(EX)'}(\mathbf{y}^{(n)})\mathbf{f}^{EX}(\mathbf{y}^{(n)}) + O(\Delta t^3)$$
$$+ \Delta t\left(\mathbf{f}^{IM}(\mathbf{y}^{(n)}) + \frac{\Delta t}{2}\mathbf{f}^{(IM)'}(\mathbf{y}^{(n)})\mathbf{f}^{IM}(\mathbf{y}^{(n)})\right) + O(\Delta t^3). \tag{B6}$$

By rearranging the terms, we obtain (B7), which matches the Taylor expansion

of the exact solution to second order.

$$y^{(n+1)} = y^{(n)} + \Delta \left( tf^{\text{EX}}\left(y^{(n)}\right) + f^{\text{IM}}\left(y^{(n)}\right) \right)$$
$$+ \frac{\Delta t^2}{2}\left( f^{(\text{EX})'}\left(y^{(n)}\right) f^{\text{EX}}\left(y^{(n)}\right) + f^{(\text{IM})'}\left(y^{(n)}\right) f^{\text{IM}}\left(y^{(n)}\right) \right) + O(\Delta t^3).$$

(B7)

This Hence, the scheme is formally second-order accurate in time, despite the use of a first-order implicit method.

---

## Author Response (AR1)

Dear editor and referees,

We appreciate the thorough and instructive comments on our manuscript from the editor and the reviewers. We have considered all the comments to incorporate changes in the revised manuscript. In the revised version, the major revisions include: (1) adding the description about cases and methods; (2) incorporating plans for model improvement and potential extensions into the conclusion section; (3) softening the second-order accuracy claim of time discretization in the manuscript.

Sincerely,

Qinghe Zhang

**REVIEWER COMMENTS**

RC1: In this paper, the authors describe their development of a novel 3D coastal and estuarine modelling system called DGCEMS based on the nodal discontinuous Galerkin method. Through some tests, it has been demonstrated that the model has second-order convergence, low spurious mixing, and capability to simulate salt-freshwater interactions in the presence of wetting and drying boundaries. The subject of the paper is well presented, and definitely of interest to the modeling community. I'd recommend the paper for publication, after addressing the following comments.

Answer: Thank you for your evaluation and suggestions. We have addressed each of your comments in the revised manuscript and made the necessary modifications accordingly.

1.In the governing equations, no specific vertical stratification was given, but Figure 1 shows 2 layers, while 10(Line 230) and 20 layers(Line 254) were used in Section 3.1 and 3.2, respectively. How is vertical stratification determined?

Answer: Thank you for your comments. The vertical layering of the model can theoretically be arbitrary. The more layers there are, the higher the accuracy of vertical flow velocity will be, but it will not increase the convergence order of the solution and will also increase the calculation time of the model. The two layers shown in Figure 1 are intended to clearly express the spatial distribution of interpolation nodes in the vertical direction. We added necessary explanations in the revised version.

**Line 138-139:**

**Figure 1**: Schematic of vertical computational element distribution (example with two layers). Black dots represent the interpolation nodes corresponding to the horizontal one-order and vertical one-order basis functions.

**Line 115-120:**

It should be noted that the number of vertical layers in the model is flexible and user-defined. The vertical discretization shown in Figure 1 employs two layers for illustrative purposes only, to demonstrate the distribution of interpolation nodes in the  $\sigma$ -coordinate direction. In practical simulations, more vertical layers (e.g., 10 to 20 layers) are typically used to improve the resolution of vertical velocity profiles and stratification. Increasing the number of layers enhances vertical accuracy but does not change the order of numerical convergence and may increase computational cost.

2. When presenting the model algorithm, it is necessary to highlight the innovative points of the solution, which can help readers better understand.

Answer: Thank you for your suggestions. We emphasized the significance of using the wet dry treatment and limiters in the mode non-split model algorithm. The 3D limiters are always applied after achieving the physical field to prevent pathological solutions, and then we can obtain the vertically averaged physical field. The WD treatment is carried out after obtaining the vertically averaged physical field to ensure the conservation of water elevation and depth-averaged momentum.

**Line 198-204:**

In the workflow, after the physical variables are computed at each time step, 3D slope limiters are applied to the reconstructed solution to suppress spurious oscillations and prevent the generation of non-physical values, particularly near steep gradients or discontinuities. These limiters ensure numerical stability. Subsequently, the physical fields are vertically averaged to derive depth-integrated variables. The WD treatment is then performed on these depth-averaged variables to accurately capture shoreline movement while preserving the conservation of water surface elevation and depth-averaged momentum. This sequential process improves robustness and physical consistency in simulations involving complex wetting and drying processes.

3.In model validation, the sources of analytical and experimental solutions should be provided first. In other words, from cases in Section 3.1 to 3.4, which ones are referenced from others and which ones are used for the first time, there should be more specific explanations.

Answer: Thank you for your advice. The artificial analytical solution in Section 3.1 is a rederivation of the analytical expression in the sigma coordinate system based on Kärnä et al. (2018). The case in Section 3.4 is inspired by the examples used in Chen et al. (2022) and conducted research using our own designed grid. We added these description in the revised manuscript.

**Line 230-231:**

The baroclinic manufactured solution test was inspired by Kärnä et al. (2018), and their original z-coordinate formulation was converted into a  $\sigma$ -coordinate framework for validation.

**Reference:**

Kärnä, T., Kramer, S.C., Mitchell, L., Ham, D.A., Piggott, M.D., and Baptista, A.M.: Thetis coastal ocean model: discontinuous Galerkin discretization for the three-dimensional

hydrostatic equations. Geosci. Model Dev., 11(11): 4359-4382. https://doi.org/10.5194/gmd-11-4359-2018, 2018.

**Line 368-369:**

This test case follows the setup described by Chen et al. (2022) with the computational mesh redefined accordingly.

**Reference:**

Chen, C., Qi, J., Liu, H., Beardsley, R., Lin, H., and Cowles, G.: A wet/dry point treatment method of FVCOM, part I: Stability experiments. J. Mar. Sci. Eng., 10(7). https://doi.org/10.3390/jmse10070896, 2022.

RC2: This paper presents a 3D modeling system based on the nodal discontinuous Galerkin (DG) method, utilizing a non-split mode framework and σ-coordinates, with implicit-explicit Runge-Kutta time stepping. The authors conduct numerical tests demonstrating second-order convergence. The study is well-structured and of clear scientific significance. I recommend publication after the authors address the following minor concern:

1. The conclusion section could be strengthened by including a few sentences discussing the potential improvements and possible future extensions or implementations of the proposed method.

Answer: Thank you for your comments. The model will be further optimized and enhanced in terms of parallel computing strategies and the overall solution framework. Future extensions will include additional functionalities and modules, such as wave–current interaction, sediment transport, and biogeochemical processes, with broader applications to realistic estuarine and coastal environments. We incorporated future plans for model improvement and potential extensions into the conclusion section.

**Line 420-424:**

In future work, the model will be further optimized in terms of its parallel computing strategy and overall solution framework to improve computational efficiency and scalability. Additional physical processes and modules, such as wave–current interactions, sediment transport, and biogeochemical dynamics, will also be incorporated to enhance the model's capability in simulating more complex and realistic estuarine and coastal systems.

CC1: The paper presents an interesting coastal and estuarine modelling system based on

discontinuous Galerkin methods. The model is three-dimensional, hydrostatic, and employs a mode-nonsplit, implicit-explicit (IMEX) Runge–Kutta time integration. It is built using a quadrature-free nodal formulation and is claimed to be second-order accurate in both space and time. The mathematical formulation and notation are clearly presented, and the numerical methods are well described. My main concern lies in the claim of second-order accuracy in time. The IMEX approach treats the vertical diffusion term implicitly, while all other terms are handled explicitly. When these two components are considered separately, the explicit scheme corresponds to a second-order Runge–Kutta method:

$$y^1 = y^n + dt f(y^n)$$

 $y^n + 1 = y^n + dt/2 (f(y^n) + f(y^1))$

In contrast, the implicit part used for vertical diffusion is essentially a first-order implicit Euler method:

$$y^n+1=y^n+dt \ f(y^n+1)$$

This means that when vertical diffusion is active, even if it is not dominant, the scheme loses second-order temporal accuracy and becomes effectively first-order in time for those terms.

Unfortunately, the numerical experiments do not adequately support the claim of second-order accuracy in time. The first test, based on a manufactured solution, excludes diffusion and only assesses spatial convergence. No temporal convergence is shown, so the second-order accuracy in time is not validated in this case. Moreover, since diffusion is excluded, the test cannot address the concern regarding the implicit treatment of vertical diffusion. The second test case, a lock-exchange with constant viscosity, primarily aims to evaluate spurious horizontal mixing and shows coherent evolution of the reference potential energy. However, it does not assess the temporal accuracy of the scheme either.

Could the authors clarify how second-order accuracy in time is ensured when vertical diffusion is included? Maybe additional numerical evidence demonstrating this, or further discussion on the temporal discretization strategy, would greatly strengthen the manuscript.

Answer: Thank you for the thoughtful comments. We now respond to the concern regarding the temporal accuracy of our IMEX time discretization, particularly in the presence of vertical diffusion treated implicitly.

As correctly pointed out, the vertical diffusion term is discretized using a backward Euler scheme, which is first-order accurate in time. However, our time integration strategy is based on a class of additive Runge–Kutta methods known as IMEXRK schemes, which allow for a second-order explicit scheme to be coupled with a first-order implicit treatment, while still maintaining overall second-order accuracy. In this second-order IMEX framework, the implicit

terms do not need to be of the same order as the explicit terms. As long as the overall scheme satisfies the conditions for second-order accuracy, or if the numerical error is dominated by the explicit terms, the method can still achieve second-order accuracy.

In another word, you may find in Eq. (27) that the implicit step does not use y^n, but y^(2) instead. Here the final physical fields results are calculated by the values from the second prediction step. Besides, the coefficients in each stage must jointly satisfy the Butcher tableau in order for the coupling of the second-order explicit and first-order implicit time discretizations to achieve overall second-order accuracy. The present study follows the second-order IMEX Runge–Kutta scheme used in the Thetis model (Kärnä et al, 2018), which provides justification for the second-order accuracy in time.

We agree with the reviewer that verifying the temporal order of accuracy through convergence tests is a meaningful effort. However, designing such test cases requires carefully eliminating the influence of spatial discretization errors and involves consideration of multiple factors. We plan to conduct a more in-depth investigation on this in future work.

CC2: Thank you very much for your thoughtful reply.

While I appreciate the clarification regarding the IMEX Runge–Kutta framework and understand that it can, in principle, achieve second-order accuracy even when the implicit part alone is first-order, I believe that such an important claim should not be left without some form of justification in the paper.

Currently, the manuscript does not provide a direct reference that formally proves the secondorder accuracy of the specific scheme employed, nor does it clearly state and verify the conditions on the coefficients that would ensure the correct order. I also consulted the Thetis paper (Kärnä et al., 2018) and, unless I overlooked something, a complete demonstration of the full second-order accuracy (including the implicit contribution) is not provided there either.

I fully understand that setting up a dedicated numerical test to verify temporal convergence with vertical diffusion can be heavy and time-consuming. As an alternative, it could perhaps be helpful to include a brief theoretical analysis in the Appendix, expanding the discrete scheme and comparing it to a Taylor series expansion. This would allow one to verify that the scheme satisfies:

$$y^{n+1} = y^{n} + (f_{ex}^{n} + f_{im}^{n}) dt + (f_{ex}^{n} + f_{im}^{n})*(f_{ex}^{n} + f_{im}^{n})' dt/2$$

Such a theoretical check could be relatively lightweight to include and would fully confirm the second-order accuracy claim without requiring heavy numerical experiments.

Thank you again for your efforts and for considering this suggestion.

Answer: Thank you for your suggestion and understanding. We will include a brief theoretical analysis of the second-order temporal accuracy in the appendix. The IMEX Runge–Kutta (IMEXRK) scheme used in this work are shown in Eq.(24)-(27):

$$\frac{\mathrm{d}y}{\mathrm{d}t} = f^{\mathrm{EX}}(y) + f^{\mathrm{IM}}(y), \tag{24}$$

$$\mathbf{y}^{(1)} = \mathbf{y}^{(n)} + \Delta t \mathbf{f}^{\mathrm{EX}} \left( \mathbf{y}^{(n)} \right), \tag{25}$$

$$\mathbf{y}^{(2)} = \mathbf{y}^{(n)} + \frac{1}{2} \Delta t \left( \mathbf{f}^{EX} \left( \mathbf{y}^{(n)} \right) + \mathbf{f}^{EX} \left( \mathbf{y}^{(1)} \right) \right), \tag{26}$$

$$\mathbf{y}^{(n+1)} = \mathbf{y}^{(2)} + \Delta t \mathbf{f}^{\text{IM}} (\mathbf{y}^{(n+1)}).$$
 (27)

For explicit term, we first expand  $f^{EX}(y^{(1)})$ :

$$\boldsymbol{f}^{\mathrm{EX}}\left(\boldsymbol{y}^{(1)}\right) = \boldsymbol{f}^{\mathrm{EX}}\left(\boldsymbol{y}^{(n)} + \Delta t \boldsymbol{f}^{\mathrm{EX}}\left(\boldsymbol{y}^{(n)}\right)\right) = \boldsymbol{f}^{\mathrm{EX}}\left(\boldsymbol{y}^{(n)}\right) + \Delta t \boldsymbol{f}^{\mathrm{(EX)'}}\left(\boldsymbol{y}^{(n)}\right) \boldsymbol{f}^{\mathrm{EX}}\left(\boldsymbol{y}^{(n)}\right) + O(\Delta t^{2}).$$
(B1)

Substituting into (26), we obtain:

$$\mathbf{y}^{(2)} = \mathbf{y}^{(n)} + \Delta t \mathbf{f}^{\mathrm{EX}} \left( \mathbf{y}^{(n)} \right) + \frac{\Delta t^2}{2} \mathbf{f}^{(\mathrm{EX})'} \left( \mathbf{y}^{(n)} \right) \mathbf{f}^{\mathrm{EX}} \left( \mathbf{y}^{(n)} \right) + O(\Delta t^3).$$
(B2)

The implicit term can be written via a fixed-point expansion:

$$\mathbf{y}^{(n+1)} = \mathbf{y}^{(2)} + \Delta t \mathbf{f}^{\text{IM}} \left( \mathbf{y}^{(2)} \right) + O\left(\Delta t^2\right). \tag{B3}$$

Substituting (B2) into (B3), we obtain:

$$\mathbf{y}^{(n+1)} = \mathbf{y}^{(n)} + \Delta t \mathbf{f}^{\mathrm{EX}} \left( \mathbf{y}^{(n)} \right) + \frac{\Delta t^{2}}{2} \mathbf{f}^{(\mathrm{EX})'} \left( \mathbf{y}^{(n)} \right) \mathbf{f}^{\mathrm{EX}} \left( \mathbf{y}^{(n)} \right) + O(\Delta t^{3})$$
$$+ \Delta t \mathbf{f}^{\mathrm{IM}} \left( \mathbf{y}^{(2)} \right) + O(\Delta t^{2}). \tag{B4}$$

Then, we expand  $f^{\text{IM}}(y^{(2)})$  and it:

$$\boldsymbol{f}^{\mathrm{IM}}\left(\boldsymbol{y}^{(2)}\right) = \boldsymbol{f}^{\mathrm{IM}}\left(\boldsymbol{y}^{(n)}\right) + \frac{\Delta t}{2} \boldsymbol{f}^{\mathrm{(IM)'}}\left(\boldsymbol{y}^{(n)}\right) \boldsymbol{f}^{\mathrm{IM}}\left(\boldsymbol{y}^{(n)}\right) + O(\Delta t^{2}). \tag{B5}$$

We can find that  $f^{\text{IM}}(y^{(2)})$  is first-order and  $\Delta t f^{\text{IM}}(y^{(2)})$  is second-order. Substituting (B5) into (B4), we obtain:

$$\mathbf{y}^{(n+1)} = \mathbf{y}^{(n)} + \Delta t \mathbf{f}^{\mathrm{EX}} \left( \mathbf{y}^{(n)} \right) + \frac{\Delta t^{2}}{2} \mathbf{f}^{(\mathrm{EX})'} \left( \mathbf{y}^{(n)} \right) \mathbf{f}^{\mathrm{EX}} \left( \mathbf{y}^{(n)} \right) + O(\Delta t^{3})$$

$$+ \Delta t \left( \mathbf{f}^{\mathrm{IM}} \left( \mathbf{y}^{(n)} \right) + \frac{\Delta t}{2} \mathbf{f}^{(\mathrm{IM})'} \left( \mathbf{y}^{(n)} \right) \mathbf{f}^{\mathrm{IM}} \left( \mathbf{y}^{(n)} \right) \right) + O(\Delta t^{3}).$$
(B6)

By rearranging the terms, we obtain (B7), which matches the Taylor expansion of the exact solution to second order.

$$y^{(n+1)} = y^{(n)} + \Delta \left( t f^{\text{EX}} \left( y^{(n)} \right) + f^{\text{IM}} \left( y^{(n)} \right) \right)$$

$$+ \frac{\Delta t^{2}}{2} \left( f^{(\text{EX})'} \left( y^{(n)} \right) f^{\text{EX}} \left( y^{(n)} \right) + f^{(\text{IM})'} \left( y^{(n)} \right) f^{\text{IM}} \left( y^{(n)} \right) \right) + O(\Delta t^{3}).$$
(B7)

Hence, the scheme is formally second-order accurate in time, despite the use of a first-order implicit method.

CC3: Thank you again for your effort in providing a theoretical analysis in the supplementary material.

However, after reviewing it carefully, I still have some concerns regarding the validity of the second-order accuracy claim. The final formula appears to be incomplete and misses cross terms required to match the full second-order Taylor expansion.

Specifically, in your expansion you arrive at:

$$y^n+1 = y^n + (f ex^n + f im^n) dt + ((f ex^n + (f ex^n')) + (f im^n + (f im^n)')) dt^2/2$$

However, this does not match the expected second-order expansion of a solution to y' = f(x) + f(y), which is:

$$y^n+1 = y^n + (f_ex^n + f_im^n) dt + (f_ex^n + f_im^n)^*(f_ex^n + f_im^n)' dt^2/2$$

$$= y^n + (f_ex^n + f_im^n) dt + ((f_ex^n + (f_ex^n)')^+(f_ex^n + (f_im^n)')^+(f_im^n + (f_ex^n)')^+(f_im^n + (f_im^n)')) dt^2/2$$

These cross terms are essential to correctly capture the interaction between the explicit and implicit parts of the system, and they are missing in your derivation.

Additionally, in Eq. (B2), you express  $y^2 = y^n + (f_ex^n) dt + (f_ex^n)^*(f_ex^n)' dt^2/2 + O(dt^3)$ . However, in Eq. (B5), when evaluating f im  $(y^2)$ , you write:

$$f_{im}(y^2) = f_{im}(y^n) + (f_{im}^n)*(f_{im}^n)' dt/2 + O(dt^2)$$

This is inconsistent. If you substitute the expression for  $y^2$  from B2 into a first-order expansion of  $f_{im}(y^2)$ , you should obtain:

$$f \text{ im } (y^2) = f \text{ im}(y^n) + (f \text{ ex}^n)^*(f \text{ im}^n)' dt + O(dt^2)$$

That is, the derivative term should be multiplied by  $f_ex(y^n)$ , and the factor 1/2 should not appear. Unfortunately, even with this correction, the final formula would remain incorrect.

In conclusion, even with the revised steps, the derivation does not convincingly establish second-order accuracy in time for the full IMEX scheme. The missing terms are not simply technicalities. They are structurally required to validate the claim.

That said, I appreciate the value of the work presented in the paper, and I recognize that this specific point is not central to the overall contribution. In light of the current state of the analysis and the absence of a temporal convergence test, it might be more appropriate to soften the second-order accuracy claim in the main text.

Thank you again for the constructive exchange and for your contributions to the modeling community.

Answer: Thank you for your guidance on this manuscript. We have re-examined the previous derivations. We think that Eq. (24) to (25) represent a complete time step with a step size of  $\Delta t$ , while Eq. (26) employs an intermediate step (essentially rewinding to  $0.5\Delta t$ ) to achieve better approximation. Thus, the expansion of Equation B5 can correspond to half a time step. Nevertheless, when cross-terms are retained, it is indeed theoretically impossible to rigorously prove that the temporal accuracy reaches second order after incorporating vertical diffusion.

Since the implicit part is computed after two explicit updates, simultaneous Taylor expansions for proving second-order temporal accuracy may not hold. The Thetis (2018) paper considered that this method achieves second-order accuracy in both time and space, which is why similar descriptions were adopted in our original text. As you noted, although the simulation results exhibit second-order convergence, a strict theoretical analysis of spatiotemporal second-order accuracy will be addressed in future studies. Accordingly, we will soften the second-order accuracy claim in the manuscript.

Thank you again for the constructive exchange, and it has been immensely valuable and greatly enriched our understanding.

According to Eqs. (24-27):

$$\frac{\mathrm{d}\mathbf{y}}{\mathrm{d}t} = \mathbf{f}^{\mathrm{EX}}(\mathbf{y}) + \mathbf{f}^{\mathrm{IM}}(\mathbf{y}), \tag{24}$$

$$\mathbf{y}^{(1)} = \mathbf{y}^{(n)} + \Delta t \mathbf{f}^{\mathrm{EX}} \left( \mathbf{y}^{(n)} \right), \tag{25}$$

$$\mathbf{y}^{(2)} = \mathbf{y}^{(n)} + \frac{1}{2} \Delta t \left( \mathbf{f}^{EX} \left( \mathbf{y}^{(n)} \right) + \mathbf{f}^{EX} \left( \mathbf{y}^{(1)} \right) \right), \tag{26}$$

$$\mathbf{y}^{(n+1)} = \mathbf{y}^{(2)} + \Delta t \mathbf{f}^{\text{IM}} \left( \mathbf{y}^{(n+1)} \right).$$
 (27)

For the explicit term, we first expand  $f^{EX}(y^{(1)})$ :

$$\boldsymbol{f}^{\mathrm{EX}}\left(\boldsymbol{y}^{(1)}\right) = \boldsymbol{f}^{\mathrm{EX}}\left(\boldsymbol{y}^{(n)} + \Delta t \boldsymbol{f}^{\mathrm{EX}}\left(\boldsymbol{y}^{(n)}\right)\right) = \boldsymbol{f}^{\mathrm{EX}}\left(\boldsymbol{y}^{(n)}\right) + \Delta t \boldsymbol{f}^{(\mathrm{EX})'}\left(\boldsymbol{y}^{(n)}\right) \boldsymbol{f}^{\mathrm{EX}}\left(\boldsymbol{y}^{(n)}\right) + O(\Delta t^{2}).$$
(B1)

Substituting into (26), we obtain:

$$\mathbf{y}^{(2)} = \mathbf{y}^{(n)} + \Delta t \mathbf{f}^{\mathrm{EX}} \left( \mathbf{y}^{(n)} \right) + \frac{\Delta t^2}{2} \mathbf{f}^{(\mathrm{EX})'} \left( \mathbf{y}^{(n)} \right) \mathbf{f}^{\mathrm{EX}} \left( \mathbf{y}^{(n)} \right) + O(\Delta t^3).$$
(B2)

The implicit term can be written via a fixed-point expansion:

$$\mathbf{y}^{(n+1)} = \mathbf{y}^{(2)} + \Delta t \mathbf{f}^{\mathrm{IM}} \left( \mathbf{y}^{(2)} \right) + O(\Delta t^2).$$
(B3)

Substituting (B2) into (B3), we obtain:

$$\mathbf{y}^{(n+1)} = \mathbf{y}^{(n)} + \Delta t \mathbf{f}^{\mathrm{EX}} \left( \mathbf{y}^{(n)} \right) + \frac{\Delta t^{2}}{2} \mathbf{f}^{(\mathrm{EX})'} \left( \mathbf{y}^{(n)} \right) \mathbf{f}^{\mathrm{EX}} \left( \mathbf{y}^{(n)} \right) + O(\Delta t^{3})$$
$$+ \Delta t \mathbf{f}^{\mathrm{IM}} \left( \mathbf{y}^{(2)} \right) + O(\Delta t^{2}). \tag{B4}$$

Then, we expand  $f^{\text{IM}}(y^{(2)})$  and it:

$$\boldsymbol{f}^{\mathrm{IM}}\left(\boldsymbol{y}^{(2)}\right) = \boldsymbol{f}^{\mathrm{IM}}\left(\boldsymbol{y}^{(n)}\right) + \frac{\Delta t}{2} \boldsymbol{f}^{\mathrm{(IM)'}}\left(\boldsymbol{y}^{(n)}\right) \boldsymbol{f}^{\mathrm{IM}}\left(\boldsymbol{y}^{(n)}\right) + O(\Delta t^{2}). \tag{B5}$$

We can find that  $f^{\text{IM}}(y^{(2)})$  is first-order and  $\Delta t f^{\text{IM}}(y^{(2)})$  is second-order. Substituting (B5) into (B4), we obtain:

$$\mathbf{y}^{(n+1)} = \mathbf{y}^{(n)} + \Delta t \mathbf{f}^{\mathrm{EX}} \left( \mathbf{y}^{(n)} \right) + \frac{\Delta t^{2}}{2} \mathbf{f}^{(\mathrm{EX})'} \left( \mathbf{y}^{(n)} \right) \mathbf{f}^{\mathrm{EX}} \left( \mathbf{y}^{(n)} \right) + O(\Delta t^{3})$$
$$+ \Delta t \left( \mathbf{f}^{\mathrm{IM}} \left( \mathbf{y}^{(n)} \right) + \Delta t \mathbf{f}^{(\mathrm{IM})'} \left( \mathbf{y}^{(n)} \right) \mathbf{f}^{\mathrm{IM}} \left( \mathbf{y}^{(n)} \right) \right) + O(\Delta t^{3}).$$
(B6)

By rearranging the terms, we obtain (B7), which matches the Taylor expansion of the exact solution to second order.

$$y^{(n+1)} = y^{(n)} + \Delta t \left( f^{\text{EX}} \left( y^{(n)} \right) + f^{\text{IM}} \left( y^{(n)} \right) \right)$$

$$+ \frac{\Delta t^{2}}{2} \left( f^{(\text{EX})'} \left( y^{(n)} \right) f^{\text{EX}} \left( y^{(n)} \right) + f^{(\text{IM})'} \left( y^{(n)} \right) f^{\text{IM}} \left( y^{(n)} \right) \right)$$

$$+ \frac{\Delta t^{2}}{2} f^{(\text{IM})'} \left( y^{(n)} \right) f^{\text{IM}} \left( y^{(n)} \right) + O(\Delta t^{3}), \tag{B7}$$

which is different from the standard Taylor expansion at the third line:

$$y^{(n+1)} = y^{(n)} + \Delta t \left( f^{\text{EX}} \left( y^{(n)} \right) + f^{\text{IM}} \left( y^{(n)} \right) \right)$$

$$+ \frac{\Delta t^{2}}{2} \left( f^{(\text{EX})'} \left( y^{(n)} \right) f^{\text{EX}} \left( y^{(n)} \right) + f^{(\text{IM})'} \left( y^{(n)} \right) f^{\text{IM}} \left( y^{(n)} \right) \right)$$

$$+ \frac{\Delta t^{2}}{2} \left( f^{(\text{EX})'} \left( y^{(n)} \right) f^{\text{IM}} \left( y^{(n)} \right) + f^{(\text{IM})'} \left( y^{(n)} \right) f^{\text{EX}} \left( y^{(n)} \right) \right) + O(\Delta t^{3}).$$
(B8)

In the revised manuscript, the description of second-order time discretization is removed.

**Line 14-15:**

The model adopts  $\sigma$ -coordinates, employs a non-split mode framework, and integrates a semi-implicit Runge–Kutta scheme.

**Line 414-415:**

A semi-implicit Runge-Kutta scheme is applied in the model.

---

## Author Response (AR2)

Dear Editor,

Thank you for your message and for the opportunity to submit our work to Geoscientific Model Development. In the revised version, the major revisions include: (1) adding the Figure captions and listing the figures; (2) using the initials instead of the full names of authors in the section "Author's contribution"; (3) giving the written-out explanation of scientific abbreviations.

Sincerely,

Qinghe Zhang

**REVIEWER COMMENTS**

1. Please note, if you used scientific abbreviations without giving the written-out explanation, these must be written out with the next file upload request. However, do not forget that there is a limit to characters (not words!) for "Short summary": it must be < 500 characters. 2. For the next revision, please use the initials instead of the full names of authors in the section "Author's contribution".

**Figure Captions**

- Figure 1: Schematic of vertical computational element distribution (example with two layers). Black dots represent the interpolation nodes corresponding to the horizontal one-order and vertical one-order basis functions.
- Figure 2: Convergence of the  $L_2$  error of (a) surface elevation, (b) horizontal velocity, (c) vertical velocity, and (d) temperature field in the baroclinic manufactured solution test case.
- Figure 3: The density in the lock exchange test under three different grid Reynolds numbers.
- Figure 4: Simulating results of (a) relative RPE and (b) the density front location at bottom layer of the lock exchange test with different Re.
- Figure 5: The topography of the ideal river plume experiment.
- Figure 6: The computational mesh domain and the initial salinity field. The salinity is 0 psu at x < 0 km and 32psu at  $x \ge 0$ km.
- Figure 7: Simulated surface river plume and surface current velocity at 48 hours. The grid resolution is refined from 1500m (left panel) to 375 m (right panel).
- Figure 8: Freshwater content at per salinity class at 48 hours' results with different grid resolutions.
- Figure 9: Surface salinity at the profile of y = 140 km under three grid resolutions. Blue and red dashed lines are salinity relative errors with grid size 1500 m and 750 m respectively compared to that of 375 m.
- Figure 10: The calculation domain and its grid division of the semi-closed estuary. The minimum grid resolution is 200 m. The river channel cross-section is symmetric about the channel centerline and uniform along the channel direction. The initial water depth is 10.2 m and  $\alpha = 0.0007$ . Three characteristic points are marked: the channel center (A), the WD boundary (B), and the initially dry location (C). The initial salinity is set to 0 on the left side of the red

solid line and 35 psu on the right side.

Figure 11: A 3D view of a flooding time.

Figure 12: Time series of surface water elevation (a), depth-averaged velocity along the channel

(b) and cross the channel (c) at points A, B, and C.

Figure 13: Salinity distribution along the channel centerline cross-section at 1.5 h, 3 h, and 6 h.

Line 11-14:

Numerical method of discontinuous Galerkin (DG) discretization for coastal ocean modelling

have advanced significantly, but there are still challenges in accurately simulating phenomena

such as wetting and drying process and baroclinic flows in coastal and estuarine regions. This

study develops a novel three-dimensional coastal and estuarine modelling system named

DGCEMS, using a quadrature-free nodal DG method.

**Author contributions**

Z.C.: Conceptualization, Methodology, Software, Validation, Writing - Original Draft,

Visualization;

Q.Z.: Conceptualization, Writing - Review & Editing, Supervision, Project administration,

Funding acquisition;

G.R.: Methodology, Software, Supervision;

Y.N.: Validation, Formal analysis, Visualization.